# Do cash transfers alleviate common mental disorders in low- and middle-income countries? A systematic review and meta-analysis

**Clara Wollburg** [1]*, **Janina Isabel Steinert** [2], **Aaron Reeves**[1], **Elizabeth Nye**[1]

**1** Department of Social Policy and Intervention, University of Oxford, Oxford, United Kingdom, **2** TUM School of Social Sciences and Technology, Technical University Munich, Munich, Germany

* mail@clarawollburg.com

**Data Availability Statement:** All relevant data are within the manuscript and its Supporting Information files.

## Abstract

A large literature has demonstrated the link between poverty and mental ill-health. Yet, the potential causal effects of poverty alleviation measures on mental disorders are not well-understood. In this systematic review, we summarize the evidence of the effects of a particular kind of poverty alleviation mechanism on mental health: the provision of cash transfers in low- and middle-income countries. We searched eleven databases and websites and assessed over 4,000 studies for eligibility. Randomized controlled trials evaluating the effects of cash transfers on depression, anxiety, and stress were included. All programs targeted adults or adolescents living in poverty. Overall, 17 studies, comprising 26,794 participants in Sub-Saharan Africa, Latin America, and South Asia, met the inclusion criteria of this review. Studies were critically appraised using Cochrane's Risk of Bias tool and publication bias was tested using funnel plots, egger's regression, and sensitivity analyses. The review was registered in PROSPERO (CRD42020186955). Meta-analysis showed that cash transfers significantly reduced depression and anxiety of recipients ($d_{pooled}$ = -0.10; 95%-CI: -0.15, -0.05; p<0.01). However, improvements may not be sustained 2–9 years after program cessation ($d_{pooled}$ = -0.05; 95%-CI: -0.14, 0.04; ns). Meta-regression indicates that impacts were larger for unconditional transfers ($d_{pooled}$ = -0.14; 95%-CI: -0.17, -0.10; p<0.01) than for conditional programs ($d_{pooled}$ = 0.10; 95%-CI: 0.07, 0.13; p<0.01). Effects on stress were insignificant and confidence intervals include both the possibility of meaningful reductions and small increases in stress ($d_{pooled}$ = -0.10; 95%-CI: -0.32, 0.12; ns). Overall, our findings suggest that cash transfers can play a role in alleviating depression and anxiety disorders. Yet, continued financial support may be necessary to enable longer-term improvements. Impacts are comparable in size to the effects of cash transfers on, e.g., children's test scores and child labor. Our findings further raise caution about potential adverse effects of conditionality on mental health, although more evidence is needed to draw robust conclusions.

**Funding:** JS was supported by the Joachim Herz Foundation (https://www.joachim-herz-stiftung.de/en/). AR received funding from the Wellcome Trust (220206/Z/20/Z, https://wellcome.org). The funders had no role in study design, data collection and analysis, decision to publish, or preparation of the manuscript.

**Competing interests:** The authors have declared that no competing interests exist.

# 1) Introduction

Common mental disorders are a critical public health issue, with more than 250 million people affected by anxiety and over 300 million people suffering from depression globally [1]. More than 80% of the global burden of common mental disorders, i.e. depression and anxiety disorders, falls on low- and middle-income countries (LMICs) [1]. People who live in poverty in LMICs are disproportionately affected by mental ill-health due to their exposure to risk factors, such as food insecurity, social exclusion, trauma and violence [2–8]. Poverty is further correlated with higher levels of stress and lower quality of life and psychosocial wellbeing [8,9]. The high prevalence of mental disorders is not only damaging in itself, but can contribute to the persistence of poverty through, e.g., lower productivity, stigmatization, higher health expenditures, and reduced self-control [8,10,11].

The association between poverty and mental health is not a spurious correlation, however, and several studies indicate that economic poverty causes and exacerbates psychological disorders [8,12]. Together, these studies support the social causation hypothesis, which states that living in conditions of poverty is connected to a multitude of severe stressors that increase vulnerability to mental disorders [6,10]. Accordingly, interventions that reduce monetary poverty may intervene on the social causation pathways and, thus, can help alleviate the burden of common mental disorders in targeted populations.

In this systematic review, therefore, we collate the causal evidence on the psychological impacts of one such intervention–the provision of cash transfers–to investigate whether monetary poverty alleviation can play a role in strengthening psychological health in LMICs.

Cash transfer programs (CTPs), which provide a financial safety net to people in poverty, are hypothesized to improve mental health outcomes by, e.g., reducing financial strain and instability, which has been linked to heightened stress and depressive symptoms in LMICs [3]. Cash transfers were also shown to facilitate social inclusion [4,13] and to improve physical health indicators [14–16], factors which have been linked to stress, anxiety and depressive symptoms in LMICs [3,6,17].

On the other hand, CTPs may adversely affect mental health, as they can aggravate social exclusion [18] and are sometimes associated with increased health risks, e.g. higher body mass index [19,20], thereby potentially increasing risks of mental ill-health. Finally, making CTPs conditional on certain behaviours can increase feelings of stress when conditions are overly demanding [21]. Conditionality may further exclude particularly vulnerable individuals, which could increase mental health inequalities [22,23].

As CTPs become a common component of social protection and poverty reduction in LMICs, it is important to better understand their effects on common mental disorders. In this systematic review and meta-analysis, we synthesize the evidence from randomized controlled trials on the mental health effects of CTPs in LMICs. Specifically, our main objectives are:

1. To analyze the effects of CTPs on depression and anxiety in LMICs, compared to inactive control groups.

2. To examine how CTPs impact recipients' stress, compared to inactive control groups.

3. To study whether unconditional and conditional CTPs differ in their effects on mental health.

4. To study the long-term psychological effects of CTPs after program cessation.

To our knowledge, this is the most comprehensive systematic review and meta-analysis on the mental health effects of cash transfers. Three earlier reviews considered studies on cash transfers and common mental disorders [11,14,24] and found some encouraging effects on children's cortisol levels and aggressive symptoms, and positive, but insignificant, effects on

depression and anxiety of children and adults [25–28]. Yet, since relevant studies published over the last decade are not included, these reviews do not provide a timely, comprehensive overview of current evidence. Moreover, none of the earlier reviews conducted a meta-analysis to offer a quantitative summary of the results because of the limited number of available studies. Three recent papers have synthesized more current evidence. Ridley et al. (2020) conducted a literature review and meta-analysis of the psychological impacts of anti-poverty programs, including cash transfers [12]. Our review extends their work by using a systematic review methodology and comprehensive search strategy, which allowed us to retrieve additional relevant studies. Moreover, our review pushes the literature forward by providing a comparison of the impacts of conditional and unconditional cash transfers on mental health. Making the receipt of transfers conditional on requirements, such as regular school attendance of children and health-clinic visits, has been shown to increase effectiveness for those outcomes [14,29]. However, adding an element of conditionality may also lead to unintended consequences, such as increased stress among recipients [21] or adverse effects for those loosing eligibility [22]. As a result, conditionality might negatively affect psychological health and could exacerbate mental health inequalities. Understanding the psychological effects of conditionality has therefore important implications for CTPs as a social protection strategy.

Secondly, Zimmerman et al. (2021) published a systematic review and meta-analysis on the effects of cash transfers on mental health outcomes of children and young people [30]. Since Zimmerman and colleagues focus on children and adolescents, however, their analysis does not include many of the studies analyzed in this review.

Finally, McGuire et al. (2020) conducted a systematic review of quasi-experimental and experimental studies of CTPs and their effects on subjective wellbeing and mental health [31]. Our review complements their work in two ways. Firstly, we were able to retrieve additional grey literature papers [32–34]. In light of the likelihood of publication bias detected in this review (see section 3.6), this is an important addition and allows us to paint a more comprehensive picture of the effects. Moreover, we use an advanced meta-analytic method–robust variance estimation (RVE)–which corrects for dependency between effect sizes, thus producing more accurate estimations of the overall effect than other meta-analytic models [35]. Moreover, RVE allowed us to integrate all reported effect sizes, thus avoiding loss of information. This is highly relevant for the cash transfer literature as studies often analyze data from the same CTP or report multiple effect sizes per sample.

## 2) Methods

This review follows the Preferred Reporting Items for Systematic Reviews and Meta-Analyses (PRISMA)-guidance [36].

### 2.1) Criteria for considering studies for this review

Table 1 provides an overview of the inclusion/exclusion criteria for this review, which are explained in more detail below.

**Participants, context, and comparison.** This review focuses on people living in poverty, who are recipients of CTPs. Since mental health disorders persisting into adulthood tend to manifest in the ages 12–24 [37], both adults and adolescents, who are CTP recipients, are included. The indirect effects of CTPs on family members of recipients, e.g. children, are not the focus of this review. Participants in inactive control groups, who received no transfers or were enrolled at a later stage (waitlist control), served as comparison group. Active control groups receiving alternative interventions were not included, as this makes causal inference about the effects of CTPs difficult.

**Table 1. Overview of inclusion and exclusion criteria.**

| | Inclusion criteria | Exclusion criteria | Rationale |
|---|---|---|---|
| **Participants** | Adults and adolescents, who live in poverty and receive cash transfers in LMICs as defined by the World Bank | Children; recipients in high-income countries; non-poor recipients | Ensures populations are comparable |
| **Comparison** | Inactive control group receiving treatment-as-usual, no treatment, treatment at a later stage (wait-list control) | Active control group receiving alternative intervention | Allows more robust causal inference about the effects of the intervention |
| **Intervention** | Unconditional and conditional CTPs in LMICs | Cash plus programs without pure cash arm; microfinance/-loans; vouchers/in-kind transfers | Ensures included interventions operate through comparable causal mechanisms |
| **Outcomes** | (1) Anxiety and depressive disorders (validated measures, e.g. CES-D, GHQ); (2) Stress (validated measures, e.g. Cohen's PSS) | Severe mental disorders, e.g. schizophrenia, severe bipolar disorder, intellectual disabilities | Effects of cash transfers likely limited to common mental disorders |
| **Timepoint** | Post-intervention and follow-up | None | Enables assessment of longer-term effects |
| **Study design** | RCTs, Cluster-RCTs | Observational, qualitative, case studies, simulations | Allows more robust causal inference about the effects of the intervention |

**Intervention.** We included conditional and unconditional CTPs targeted at households living in poverty in LMICs. We did not apply an absolute low-income/poverty threshold but relied on the relative threshold for grant eligibility applied by the organizations administering CTPs. We included programs providing regular small transfers as well as one-time lump-sum payments. "Cash plus"-programs, which combine CTPs with complementary interventions, e.g. antidepressant treatment, were not included in the review, as it is difficult to disentangle the causal effects of CTPs on mental health. Moreover, other economic strengthening programs, such as microloans, are not included, as they may operate through different causal mechanisms.

**Outcomes.** Our review focuses on common mental disorders, i.e. depressive and anxiety disorders [1], as well as stress. We focus on common mental disorders, as opposed to severe mental illnesses, such as schizophrenia, since socio-economic risk factors are of limited importance and, thus, CPTs may not readily impact more severe mental disorders [11,38]. Studies using validated surveys were included in this review, using e.g. the Center for Epidemiologic Studies Depression Scale, the General Health Questionnaire, or Cohen's perceived stress scale. We included both post-intervention and follow-up measurements, as they can provide important insights into longer-term effects.

**Study types.** Randomized controlled trials (RCT) and cluster-RCTs were included in this review. Due to the random allocation of participants/clusters into control and treatment units, these designs tend to ensure that known and unknown confounders are balanced between groups [39]. This reduces the risk of selection bias and allows for robust causal assumptions about the effectiveness of CTPs [39–41]. Qualitative or observational studies, which tend to have lower robustness in determining causality, were not included [39].

**Language and publication date.** Since this review is the first attempt to systematically search and synthesize the CTPs-mental health literature, we did not restrict eligibility based on studies' publication dates. Studies in English, German, and Spanish were screened for eligibility.

## 2.2) Search methods for identification of studies

We searched the following electronic databases covering relevant literature from Public Health, Psychology, Economics, and other Social Sciences: Web of Science (Clarivate), Scopus

(Elsevier), MEDLINE (Ovid), EMBASE (Ovid), PsycINFO (Ovid), Global Health (Ovid), and Econlit (ProQuest). To avoid publication bias, we further conducted grey literature searches [42]. We hand-searched the 3ie evidence portal and J-PAL database of randomized evaluations to retrieve unpublished impact evaluations and the website of The Transfer Project, the Innovations for Poverty Action database, and Ideas/Repec [43]. The last search was conducted on 17 April 2020.

The search syntax used for the literature search can be found in S1 Appendix in S1 File. The search syntax as well as the selection of databases/websites was informed by relevant literature and systematic reviews [11,29,44], the Effective Practice and Organisation of Care (EPOC) collection of databases and search terms for LMICs [43,45], and input from the University's librarian. We piloted the search terms in two databases and refined the syntax together with the librarian to achieve sufficient sensitivity [41]. No study-design filters were applied, since they tend to increase the risk of excluding relevant studies that do not use standardized methodological reporting [46]. Moreover, we included Medical Subject Heading (MeSh)-terms and used truncation and wildcards [41].

## 2.3) Data collection and analysis

**Data extraction.** Using the online-tool Rayyan, CW removed duplicate articles and screened abstracts/titles of the remaining records [47]. To reduce bias in study selection [48], JS screened 10% of titles/abstracts, which were randomly selected by searching the first 17 studies of each letter of the alphabet (excluding Q, X, and Y). Reviewer agreement was high (408 of 427 double-screened studies) and discrepancies were resolved through discussion. 184 studies were included for full-text screening, 10% of which, selected through a random number generator, were again double-screened by a second reviewer. Conflicts (1 of 18 studies) were resolved through discussion.

Results and characteristics of included studies (participant and program characteristics, setting, outcome measurement, and methodological details) were extracted in Excel guided by the EPOC data extraction form [49].

**Risk of bias.** When potential biases and study limitations are not taken into account, the results of systematic reviews can be misleading [50]. We therefore evaluated the risk of bias of included studies using the newly-updated Cochrane risk-of-bias tool (RoB-2). RoB-2 is a comprehensive analysis tool that covers biases arising due to the randomization process, deviations from intended interventions, missing outcome data, measurement of the outcome, and selection of reported results [51]. Unlike other tools, the RoB-2 employs explicit decision-making algorithms that facilitate a standardized RoB-assessment [51]. However, we decided not to apply an overall risk of bias rating per study determined by the most serious RoB, since this would inappropriately give equal weight to all domains and would not allow us to discriminate between studies with one and multiple high-risk domains [52]. Cluster-RCTs were assessed with the adapted RoB-2 for cluster-randomized trials [53], which includes an additional domain to address identification/recruitment bias that can arise in cluster-RCTs [54]. A second reviewer conducted RoB-assessments of a randomly-chosen 10% of included studies and came to the same overall RoB-judgments. Discrepancies in the answers to sub-questions per domain were resolved through discussion.

**Meta-analysis.** Meta-analysis can increase confidence in intervention effects by aggregating information from several studies in different settings, yielding higher statistical power than individual studies, and highlighting potential inconsistencies between studies [50].

Traditional fixed-effect meta-analysis relies on the assumption that primary studies are statistically independent, and their effect sizes estimate the same 'true' population effect [35,55].

In the context of social intervention research, the assumptions of the traditional fixed-effect meta-analysis model are often violated [56]. We therefore extended the model in two ways. Firstly, our analysis relies on a random-effects model, which does not assume that all studies represent variations of the same intervention effect [41]. This is a more accurate model for our data, since included studies vary in program characteristics, target groups, and country context [55]. The random-effects model may be written as:

$$y_j = \beta_0 + u_j + e_j$$

Where $y_j$ is the estimated effect in the jth study, $u_j$ represents the study-level random variation, and $e_j$ is the estimation error between the observed and 'true' effect for study j [56]. Thus, the model includes two sources of variation: between-studies ($u_j$) and within-study ($e_j$) [55].

We further adjusted the model for dependency within and between studies using robust variance estimation (RVE). RVE is an advanced meta-analytic technique that corrects standard errors of dependent effect sizes, thus producing more accurate estimations of the overall effect [35]. Studies in this review are dependent in two important ways: five trials analyze overlapping samples of the same CTP and, thus, their error terms are correlated [56]. Moreover, four trials report multiple relevant effect estimates, which are statistically dependent [35]. RVE has the advantage of accommodating both sources of dependency, which avoids loss of information as all relevant effect sizes can be included [56,57]. Moreover, RVE does not require accurate knowledge of the underlying covariance structures between effect sizes, as opposed to other meta-analytic models, e.g. multivariate meta-analysis (see [35] for demonstration). The RVE meta-analytic model used in this review may be written as:

$$y_{ij} = \beta_0 + u_j + e_{ij}$$

Where $y_{ij}$ is the ith effect size in the jth cash transfer program, $\beta_0$ represents the average population effect, $u_j$ denotes the study-level random effect, and $e_{ij}$ is the residual error term [56].

RVE was conducted in R (version 3.5.2) using the *robumeta* package [57]. We extracted effect estimates from primary studies and converted them into a common standardized effect size, Cohen's d, using Wilson's effect size calculator [58]. Inverse variance weights for correlated effects were used, so that more precise effect estimates contribute more to the overall estimate [41,56]. Following Tanner-Smith et al. (2016), we assumed a within-study/program correlation of rho = 0.8 and conducted sensitivity analyses for different rho-values. Adjustments for small sample sizes were applied and results are reported with 95%-confidence intervals [41,56]. All R code is provided in S4 Appendix in S1 File.

## Meta-regression

To explore the relative effects of conditional vs unconditional CTPs, we conducted RVE meta-regression with a mixed-effects model so as to accommodate within-study and between-study variation. The meta-regression model can be written as:

$$y_{ij} = \beta_0 + \beta_1(conditionality)_{ij} + u_j + e_{ij}$$

Where the covariate $\beta_1$, which represents a dummy variable specifying whether the CTP was conditional, is added to the RVE-model in above section [56]. Meta-regression was performed for depression and anxiety only, since all studies analyzing stress outcomes examined unconditional CTPs.

**Assessment of heterogeneity.**  A high variability across studies can produce misleading results, when it masks systematic differences between studies [46]. We therefore assessed

heterogeneity using Tau$^2$, which measures the between-study variance, and I$^2$, which estimates the percentage of between-study variability that is a result of heterogeneity rather than random error [50]. Generally, heterogeneity is considered insubstantial for I$^2$ up to 40%, moderate between 30%–60%, substantial between 50%–90%, and high between 75%–100% [50]. However, since meta-analysis with a small number of studies can lead to both under- and overestimation of heterogeneity, these categories are considered suggestive, not definite [59]. Finally, since heterogeneity estimates in RVE can be inaccurate [60], we conducted multilevel random-effects meta-analyses for all outcomes as an additional test of heterogeneity.

**Publication bias.**    Publication bias can lead to overestimation of effects in systematic reviews [61–64]. Where possible, we assessed publication bias using contour-enhanced funnel plots, which map effect sizes against standard errors [65]. Since this requires a minimum of ten studies per meta-analysis, we could only draw plots for depression and anxiety outcomes [65]. To guide the interpretation of funnel plots, we tested for significant asymmetry using Egger's regression test [66]. It should be noted that funnel plot asymmetry may occur for reasons other than publication bias and traditional methods may not adequately account for effect size dependency [65,67]. We therefore conducted further analysis using the R package *PublicationBias*, which can be applied to data with dependent effect sizes and permits an assessment of the potential impact of different degrees of publication bias on results [68].

**Confidence in the evidence.**    We assessed the certainty in the body of evidence using the Grading of Recommendations, Assessment, Development and Evaluation (GRADE) approach [69]. GRADE ranks confidence in findings from high to very low based on risk of bias, effect consistency, imprecision, indirectness, and publication bias [42].

**Robustness.**    We tested the robustness of RVE-results to different within-study/program correlation coefficients [56] and conducted additional assessments of heterogeneity using multilevel meta-analysis (MLMA) [70]. The relatively small number of studies included in meta-analysis did not permit a sensitivity analysis based on risk of bias (RoB). Excluding high-risk studies from meta-analysis could have led to underpowered, and potentially misleading, results. Instead, the impact of RoB is accounted for using the GRADE approach.

### 2.4) Difference between review and protocol

A review protocol was registered with PROSPERO (registration number CRD42020186955) and the following modification was made to the pre-specified methodology: We originally intended to conduct additional meta-regression to assess the effects of RoB and explore heterogeneity along the lines of covariates such as age and gender. However, the number of retrieved studies did not allow for this additional analysis.

## 3) Results

### 3.1) Description of the studies

5,925 records were retrieved through electronic databases and 271 through searching grey literature websites. After removing duplicates, a total of 4,143 titles/abstracts was screened, of which 184 were included for full-text review. After assessing full texts for eligibility, 17 records were included and 167 studies excluded. 109 of these did not report relevant outcomes, but focused on e.g. subjective wellbeing, 25 studies were not an RCT/cluster-RCT, 17 trials were not a CTP according to the eligibility criteria of this review, and 10 studies focused on the wrong population, e.g. children. Six studies were ongoing trials, replications, or working paper versions of included studies, or did not report relevant outcomes. S2 Appendix in S1 File lists all full-text reviewed studies and reasons for exclusion. An overview of the search process is provided in Fig 1 [36].

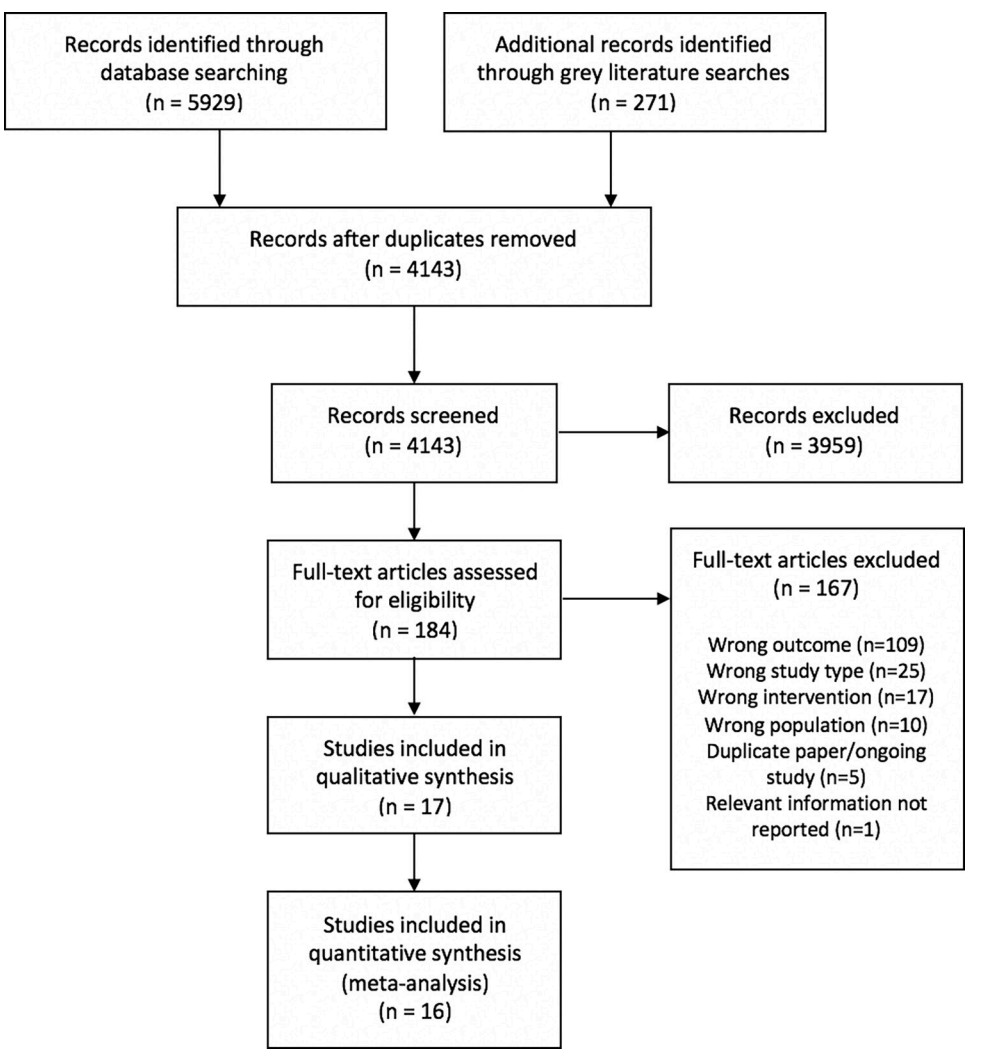

**Fig 1. PRISMA-flowchart of search process [36].**

**3.1.1) Characteristics of included studies.**   S3 Appendix in S1 File provides an overview of the characteristics of included studies, which are discussed in more detail below.

*Study type and methods.* All but three included studies were cluster-RCTs. Twelve studies were published in peer-reviewed journals, two studies were working papers [34,71], two were evaluation reports [33,72] and one record was a conference proceeding [32]. Almost all included studies reported effect sizes as regression coefficients (13/17), with four exceptions: Hjelm et al. (2017) calculated mean differences, and Kilburn et al. (2016), Pettifor, Wang et al. (2016), and Abdoulayi et al. (2017) reported binary proportions.

*Setting.* A large share of included trials was conducted in Sub-Saharan Africa (13/17), with four studies located in Malawi, four in Kenya, two in South Africa, and one each in Zambia, Mali, and Uganda. Three trials were run in Latin America (Nicaragua and Ecuador) and one in Bangladesh. With the exception of the Urban Microinsurance project [73], all trials were either conducted in rural areas (13 of 17 studies) or focused on both urban and rural households (3 of 17 studies).

*Intervention.* The 17 studies examined 13 different programs. More than half of the studies examined unconditional CTPs (10 studies), five looked at conditional programs (CCTs) and two studies compared the effects of conditional and unconditional transfers [21,72].

Conditions focused on school attendance (6 studies), health check-ups (2 studies), and/or the development of a business proposal (3 studies). Even in the absence of explicit conditionality, however, there was strong messaging around desired behaviors, i.e. investments in education and basic needs, in two studies [33,74] and in one study participants perceived transfers to be conditional even though they were not [75]. Conversely, in CCTs, conditions were not always strictly monitored [76,77].

Transfer values were equivalent varied between US$144 and US$1,525 over the length of the program, which is equivalent to approximately 9%–23% of participants monthly consumption (calculations based on 15 studies for which data was available, see S3 Appendix in S1 File for cash values per program). Most CTPs provided payments on a monthly or bimonthly basis over a period of nine months to four years. Three programs delivered one-time lump sums: GiveDirectly's CTP, Uganda's Youth Opportunity Program, and the Nairobi Urban Microinsurance Project.

*Participants and comparison groups*. Overall, the studies included 26,794 participants, with a sample size for relevant outcomes between 487 and 16,133 participants. Transfers were targeted to low-income and/or deprived communities, as indicated by, e.g., low monthly household expenditure and consumptions [34,75,76,78–80], inability to meet basic needs [33,74], food insecurity [75,81], low educational attainment and high HIV risk [21,72]. Beneficiaries were predominantly female, with eight studies targeting women/mostly women. Six studies included mixed gender samples and two studies measured mental health outcomes of male partners of female beneficiaries. More than half of the trials assessed adult mental health (13 studies), eight studies included young adults and adolescents in the ages of 13–24. Two programs–the Malawi Social Cash Transfer Program and the HIV Prevention Trials Network study–provided cash allowances to households with comorbidities, e.g. HIV and disabilities. As per inclusion criteria of this review, control groups received no transfers (11 studies) or entered the program at a later stage (6 studies). Typically, control groups lived in geographically separated villages/communes, although three studies also included 'spillover controls', i.e. households living in treatment villages but not receiving transfers [21,71,82].

*Outcomes*. 13 studies measured depression, using the 10-item or 20-item Center for Epidemiologic Studies Depression Scale (CES-D), the 9-item Patient Health Questionnaire (PHQ-9), and/or the 10-item Children's Depression Index (CDI). Two studies measured anxiety and depressive symptoms using the 12-item General Health Questionnaire (GHQ-12), and one study assessed anxiety using the Revised Children's Manifest Anxiety Scale (CMAS). Stress was measured in 11 studies, ten of which employed the 4-item or 14-item Cohen's Perceived Stress Scale (PSS). Hjelm et al. (2017) used the six negatively-worded items of Cohen's PSS, since they showed higher internal reliability.

Most studies provided post-intervention measurements (13 studies), and five studies included follow-ups at 2, 4 or 9 years after the end of the program. Data collection occurred mostly in private interviews with enumerators of the same sex or through computer/audio-assisted self-interviews (8 studies).

*Funding*. All articles disclosed sources of research funding. In 12 trials, there was partial overlap between research and program funders, which potentially introduces funding bias, i.e. reporting of results that favor the interests of funders [83,84]. Although funding bias cannot be ruled out, note that 4 out of 5 independently-funded studies reported positive results, vis-à-vis 5 out of 12 studies with overlap between research and program funders.

## 3.2) Risk of bias in included studies

As shown in Fig 2 [85], six included studies were at high risk of bias in at least one domain. This was due to significant baseline imbalances, which likely affected mental health outcomes

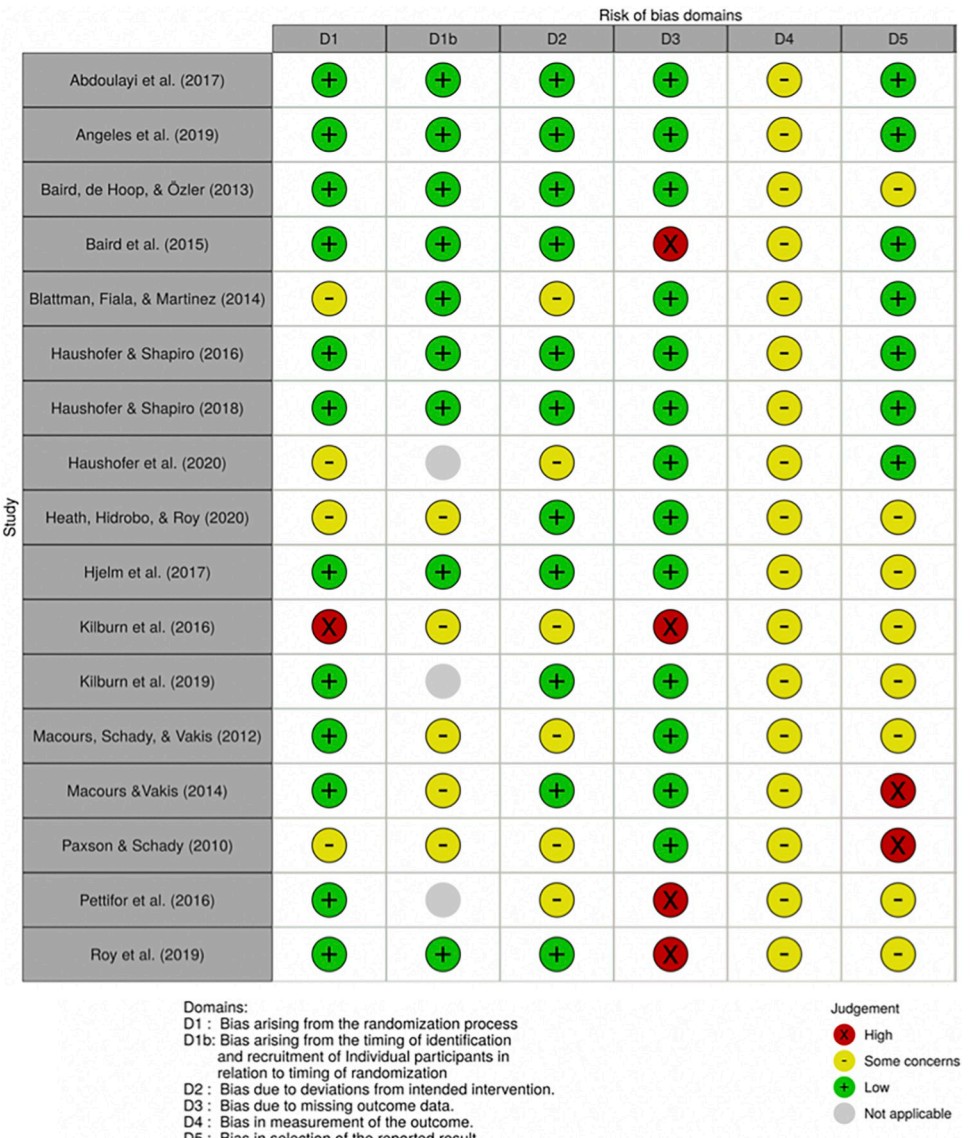

**Fig 2. Risk of bias per study.**

[78], selection bias or differential attrition that was not corrected for [34,72,78] and missing information about loss to follow-up [32]. Two studies received a high RoB-rating due to selective outcome reporting: Macours and Vakis (2014) reported depression outcomes only for one treatment arm; Paxson and Schady (2010) did not find positive effects for urban households, possibly due to implementation failure, and subsequently excluded these from the analysis. All studies used validated self-report questionnaires to measure mental health outcomes. Since participants and personnel are aware of treatment status, this could potentially influence the results, e.g. due to social desirability bias or interviewer effects [86]. Thus, following Higgins et al. (2019), all included studies are judged as raising some concerns. Moreover, eight studies raised some concerns of bias because no protocol was available or reported outcomes had not been pre-specified.

### 3.3) Effects of the intervention

**Depression and anxiety.**    11 studies reported depression outcomes, using CES-D, PHQ-9 and CDI-questionnaires, one study measured anxiety using the CMAS-questionnaire, and two studies measured common mental disorders, including depressive/anxiety symptoms, using the GHQ-12. Since CES-D, PHQ, and GHQ-measures have considerable overlap and exhibit similar response patters [87], it was concluded that anxiety and depression represent overlapping constructs. For instance, GHQ-12, CES-D and PHQ-9 all include items on sleep disturbances, feelings of depression, self-worth, or ability to concentrate [88–90]. Thus, anxiety and depression outcomes were meta-analyzed jointly.

Outcomes were grouped into two timepoints: post-intervention effects, which are measured when the program is still ongoing or has recently ended, and follow-up effects, representing outcomes measured 2–9 years after program cessation.

11 studies covering eight CTPs measured depression/anxiety post-intervention. The study by Roy et al. (2019) was not included as it only reported aggregated measures of mental health. Meta-analysis shows a significant positive, but small, effect on depression and anxiety ($d_{pooled}$ = -0.102; 95%-CI: -0.151, -0.0526; $p<0.01$). The between-study variance is $Tau^2$ = 0.0035, and the non-random across-study variation is $I^2$ = 50.11%, which indicates moderate to substantial heterogeneity. Note, however, that MLMA-estimates for heterogeneity are much smaller ($I^2$ = 15%), although statistically significant. Fig 3 illustrates that while most effects represent improvements, they tend to be small and a few confidence intervals cross zero, representing non-significant effects.

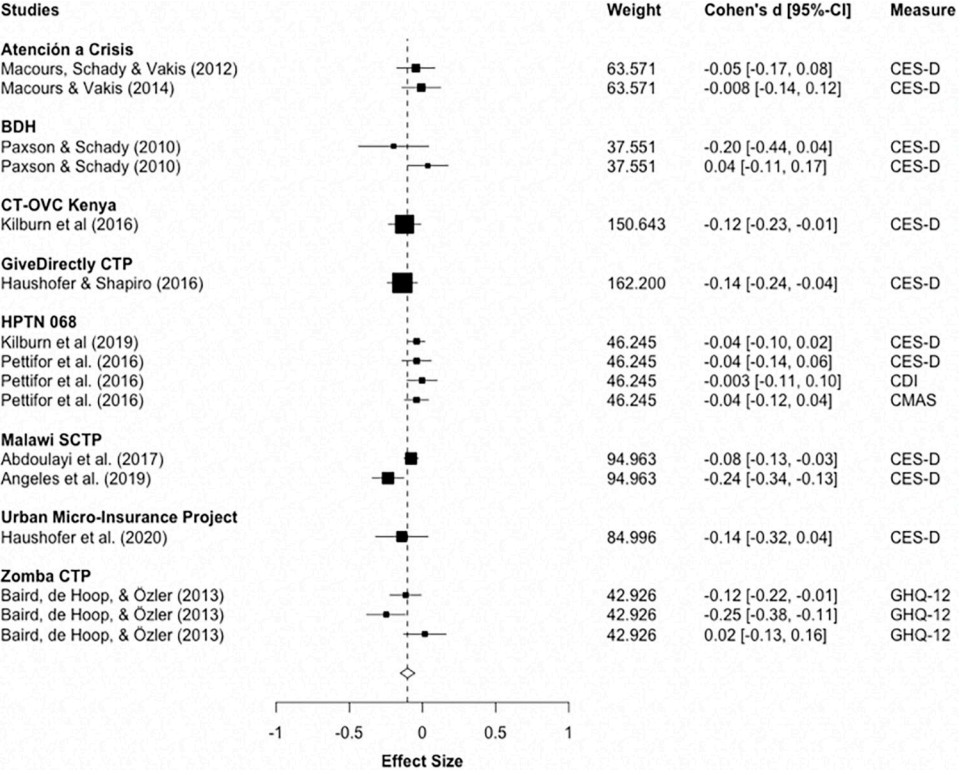

**Fig 3. Depression and anxiety post-intervention.** Notes: RVE correlated effects model with small-sample corrections. Robust standard errors adjusted at program-level. Outcomes are coded so that negative Cohen's d-values indicate improvements. Diamond and dashed line display overall estimate of $d_{pooled}$ = -0.102, width of diamond represents 95%-CI [-0.15,-0.05].

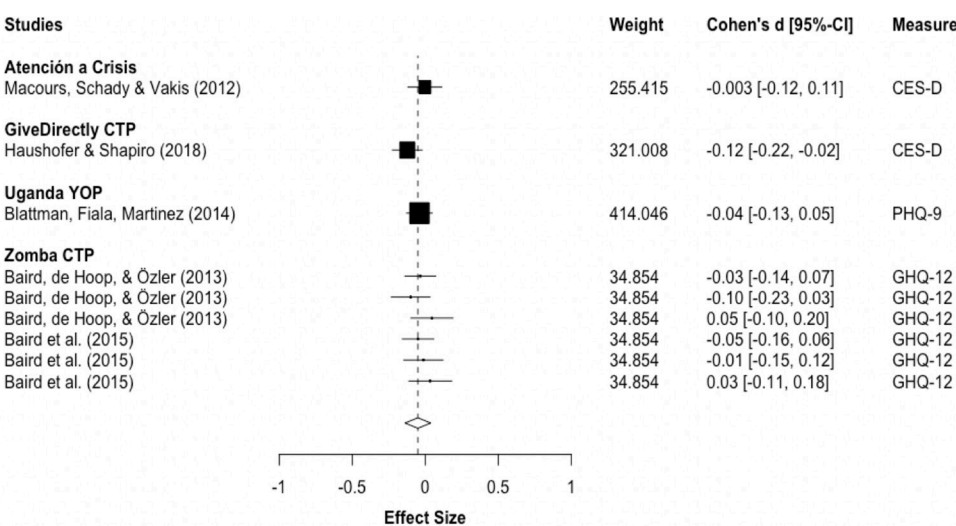

**Fig 4. Depression and anxiety at follow-up.** Notes: RVE correlated effects model with small-sample corrections. Robust standard errors adjusted at program-level. Outcomes are coded so that negative Cohen's d-values indicate improvements. Diamond and dashed line display overall estimate of $d_{pooled}$ = -0.0508, width of diamond represents 95%-CI [-0.14, 0.04].

Meta-analysis for follow-up measurements comprises five studies. RVE-estimates fail to reach significance, indicating that small improvements post-intervention may not last after program cessation ($d_{pooled}$ = -0.0508; 95%-CI: -0.139, 0.0374; ns). However, since the degrees of freedom are below 4, indicating low power, $d_{pooled}$ should be interpreted with caution. Fig 4 illustrates that effect sizes are small and several cross the line-of-no-effect. Heterogeneity is relatively low with $I^2$ = 12.15% and $Tau^2$ = 0.000412, although meta-analysis with a small number of studies may not yield accurate heterogeneity-estimates [59].

**Perceived stress.** Ten studies, which reported impacts of CTPs on recipients' stress were meta-analyzed, nine of which used Cohen's PSS, and one used all negatively-worded items of the PSS [75]. As for the analyses above, we partitioned outcomes into post-intervention effects and follow-up measurements, which were reported by two studies, 2 and 9 years after program cessation.

Seven studies, analyzing eight programs, measured recipients' stress post-intervention. RVE meta-analysis finds insignificant results with wide confidence intervals, including both the possibility of a negligible increase and small decrease in stress ($d_{pooled}$ = -0.102; 95%-CI: -0.321, 0.117; ns). Moreover, heterogeneity is considerable, with non-random across-study variability of $I^2$ = 94.03% and a $Tau^2$ value of 0.051856. Thus, $d_{pooled}$ may be misleading and should be cautiously interpreted. Fig 5 illustrates these varied effects, with some studies reporting significant, small increases in stress [28] and others finding small to moderate stress reductions [74,82].

Due to the small number of studies reporting follow-up data on perceived stress, no meta-analysis was conducted. Haushofer and Shapiro (2018) find that the moderate improvements in recipients' stress levels post-intervention, greatly decreased one year later (d = -0.12, 95%-CI: -0.22, -0.02). Blattman, Fiala and Martinez (2018) obtain insignificant effects 9 years after participants received a cash grant of approximately $400/recipient (d = -0.02, 95%-CI: -0.11, 0.07). Similar to depression/anxiety outcomes, this may indicate that CTPs do not impact stress in the longer-term. Table 2 provides an overview of RVE-results for all three meta-analysis.

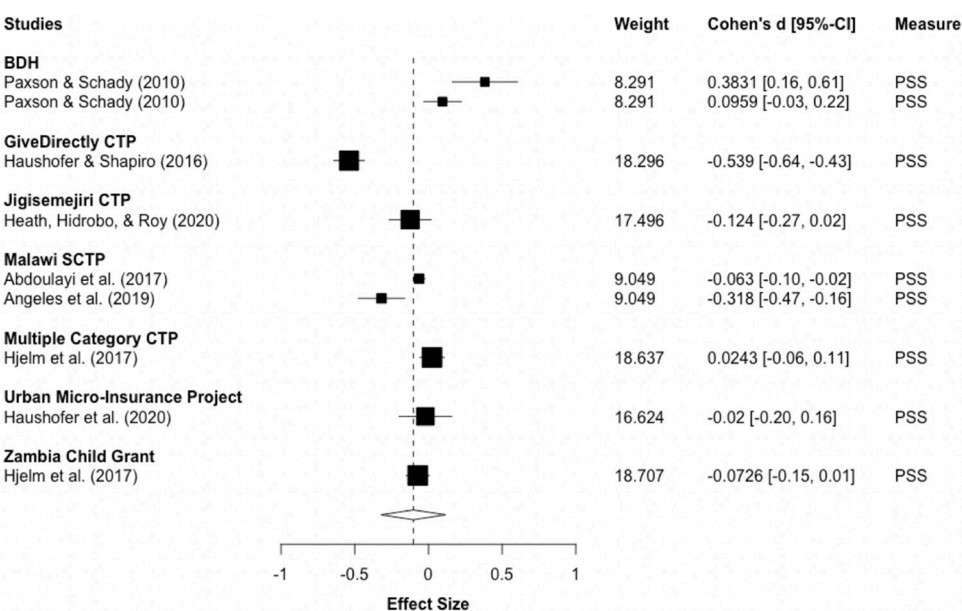

**Fig 5. Stress post-intervention.** Notes: RVE correlated effects model with small-sample corrections. Robust standard errors adjusted at program-level. Outcomes are coded so that negative Cohen's d-values indicate improvements. Diamond and dashed line display overall estimate of $d_{pooled}$ = -0.102, width of diamond represents 95%-CI [-0.32, 0.12].

## 3.4) Meta-regression

Table 3 displays the results for meta-regression, which assesses whether effects differ for conditional and unconditional programs. Meta-regression was performed for depression/anxiety outcomes only, since too few studies reported recipients' stress. To increase power, both post-intervention and follow-up measurements are included. Nonetheless, with 13 studies and 22 effect sizes, meta-regression is still relatively underpowered, as illustrated by the low df-values, and should therefore be interpreted as exploratory only. The results indicate that there are small improvements in depression/ anxiety outcomes in UCT-programs, which are larger than the effects of unconditional and conditional programs combined ($d_{pooled}$ = -0.14; 95%-CI: -0.17, -0.10; $p<0.01$). The CCT-estimate suggests that adding conditionality may decrease positive effects on depression and anxiety ($d_{pooled}$ = 0.10; 95%-CI: 0.07, 0.13; $p<0.01$). Heterogeneity was reduced to $I^2$ = 0.00%, although I-squared values may not be accurate due to the small number of studies analyzed [59].

## 3.5) Robustness

All meta-analyses were robust to different rho-values (S5 Appendix in S1 File). Conducting MLMA produced similarly small improvements in depression/anxiety post-intervention and

**Table 2. RVE meta-analysis results.**

| | Estimate | Std. Error | I-squared | 95%-CI | Df |
|---|---|---|---|---|---|
| **Anxiety & Depression** post-intervention | -0.102*** | 0.0205 | 50.11% | -0.15, -0.05 | 6.4 |
| **Anxiety & Depression** follow-up | -0.0508 | 0.0265 | 12.15% | -0.14, 0.04 | 2.78 |
| **Stress** post-intervention | -0.102 | 0.0895 | 94.03% | -0.32, 0.12 | 5.99 |

Notes: RVE correlated effects models with small-sample corrections. Robust standard errors adjusted at program-level. Df<4 indicates low power. Significance codes: < .01*** < .05** < .10*.

**Table 3. RVE meta-regression results.**

|  | Estimate | Std. Error | P-value | 95%-CI | Df |
|---|---|---|---|---|---|
| **Intercept** | -0.135*** | 0.0109 | 0.000820 | -0.17, -0.10 | 3.20 |
| **CCT** | 0.103*** | 0.0117 | 0.0000945 | 0.07, 0.13 | 6.27 |

Notes: Intercept represents $d_{pooled}$ for unconditional CTPs, CCT represents additional effect of conditionality. Df<4 indicates low power. Significance codes: < .01*** < .05** < .10*.

null-results with wide confidence intervals for perceived stress (Table 4). For depression/anxiety at follow-up, however, MLMA revealed very small significant effects, indicating that low power in the RVE-analysis may have prevented the detection of significant results. Estimates for I-squared were lower than RVE-estimates, with small between study heterogeneity for depression/anxiety at both post-intervention ($I^2$ = 11.49%) and follow-up ($I^2$ = 0.00%), but considerable heterogeneity for stress ($I^2$ = 85.74%). In line with these results, tests for heterogeneity were significant for both depression/anxiety and stress post-intervention, but insignificant for depression/anxiety at follow-up.

### 3.6) Publication bias

Fig 6 displays the contour-enhanced funnel plot for depression/anxiety post-intervention. The plot shows some asymmetry, with studies missing in areas where effects would be insignificant (white area) and favor the control rather than the intervention (right-hand side). This indicates that publication bias is plausible [65], although Egger's regression test shows no significant asymmetry (z = -0.7849, p = 0.4325). Further analysis following Mathur and Vanderweele (2019) suggests that results are relatively robust to different levels of publication bias. Assuming a 40% higher likelihood of positive findings to be published [63], meta-analysis estimates remain significant at the 5%-level (p = 0.044), although the pooled estimate would be much smaller (d = -0.036, 95%-CI: -0.07, -0.002).

## 4) Discussion

### 4.1) Discussion of the main results and confidence in the evidence

Can cash transfers play a role in alleviating common mental disorders in LMICs? This systematic review summarized the evidence on the effects of CTPs on stress, anxiety, and depressive symptoms in adults and adolescents who live in poverty. Meta-analysis of 11 studies and pooling data from 22,488 participants shows that cash grants significantly improve depression and anxiety post-intervention. Following GRADE-criteria, the overall confidence is judged to be moderate, due to some indication of publication bias and since meta-analyzed trials carried intermediate to high risk of bias. The magnitude of the effect is comparable to meta-analysis on other distal outcomes in the cash transfer literature, e.g. on child labor [91] and test scores

**Table 4. Multilevel meta-analysis results.**

|  | Estimate | Std. Error | 95%-CI | $I^2$ study | $I^2$ program | Q |
|---|---|---|---|---|---|---|
| **Depression/ Anxiety** post-intervention | -0.0835*** | 0.0197 | -0.12, -0.04 | 11.49% | 15.30% | 29.25** |
| **Depression/ Anxiety** follow-up | -0.0399* | 0.0195 | -0.08, -0.002 | 0.00% | 0.00% | 6.28 |
| **Stress** post-intervention | -0.1292 | 0.0884 | -0.30, 0.04 | 85.74% | 5.51% | 115.80*** |

Notes: Random effects multilevel model. Standard errors are adjusted for dependency at program and study level. Significance codes: < .01*** < .05** < .10*.

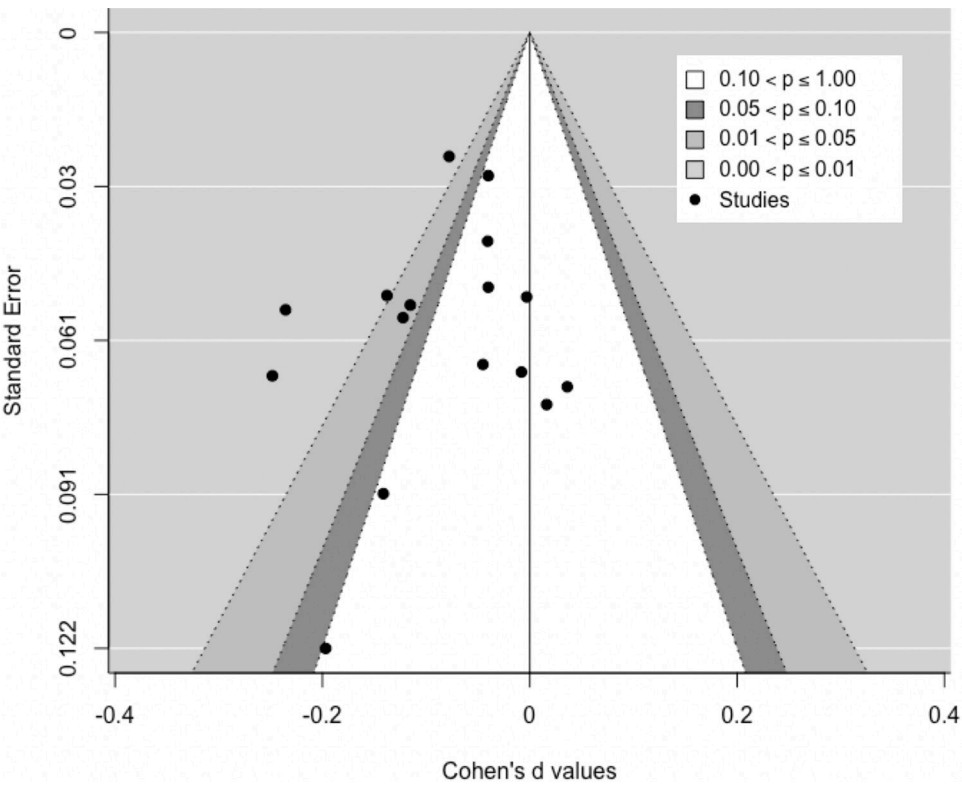

**Fig 6. Contour-enhanced funnel plot: Depression/anxiety post-intervention.**

[29]. Moreover, our effect sizes are similar to those found by Ridley et al. [12] and McGuire et al. [31], who also report significant positive impacts on common mental disorders. Zimmerman et al. [35], on the other hand, find positive, but much smaller and insignificant aggregate effects on depressive symptoms in children and young people. The authors note, however, that the validity of the pooled effect size is limited due to substantial heterogeneity.

The effects of CTPs on stress are less clear. Meta-analysis of seven studies and 19,071 recipients yielded non-significant results with wide confidence intervals including the possibility of both meaningful reductions and small increases in stress. Due to high heterogeneity and imprecision, the overall confidence is judged to be very low, and, thus, the pooled estimate may not be informative with regard to the actual effects of CTPs on stress [42]. The limited number of retrieved studies precluded further investigation of the reasons for the heterogenous results.

After program cessation, no statistically significant impact on depression and anxiety was found, although meta-analysis was underpowered, which could have prevented the detection of a significant effect. Conclusions about long-term effects are further complicated by difficulties inherent to follow-up studies, e.g. due to high attrition and mobility of participants [92]. In light of low power, relatively wide confidence intervals, and since publication bias could not be assessed, the overall certainty in depression/anxiety outcomes at follow-up was rated as low, meaning that further research may substantially change the pooled estimate of this meta-analysis [69].

There is some indication that stress effects also fade after program cessation, with two studies finding small impacts two years [71] and null-results nine years [93] after completion of the CTP.

Table 5 provides an overview of the GRADE assessment.

**Table 5. Summary of findings table & GRADE-assessment.**

| | | Certainty assessment | | | | Summary of findings | | | | | Overall certainty |
|---|---|---|---|---|---|---|---|---|---|---|---|
| Outcomes | Study design[a] | Risk of bias | Inconsistency | Indirectness | Imprecision | Publication bias | # of studies | Sample size | | Effect | |
| | | | | | | | | Treatment | Control | $d_{pooled}$ (95%-CI) | |
| **Depression & Anxiety** post-intervention | RCTs | serious[b] | not serious[e] | not serious[h] | not serious[i] | serious[l] | 11 | 11117 | 11371 | -0.10 (-0.15, -0.05)*** | ⊕⊕⊕◯ MODERATE |
| **Depression & Anxiety** follow-up | RCTs | serious[c] | not serious[f] | not serious[h] | serious[j] | NA[m] | 5 | 3390 | 4040 | -0.05 (-0.14, 0.04) | ⊕⊕◯◯ LOW |
| **Stress** post-intervention | RCTs | serious[d] | very serious[g] | not serious[h] | very serious[k] | NA[m] | 7 | 9836 | 9235 | -0.10 (-0.32, 0.12) | ⊕◯◯◯ VERY LOW |

a. Following Schünemann et al. (2013) [42], the initial confidence level placed in evidence derived from RCTs is high.

b. 4/11 studies were judged at high risk of bias due to baseline imbalances (Kilburn et al., 2016), selective reporting (Marcours & Vakis, 2014 [77]; Paxson & Schady, 2010 [28]) and missing information on attrition [32], all others raised some concerns due to self-reported outcomes.

c. 1/5 studies was at high risk of bias due to baseline imbalances (Baird et al., 2015 [72]), all others raised some concerns due to self-reported outcomes; not downgraded since high risk paper did not contribute to a large extent in overall effect.

d. 1/7 studies was a high risk of bias due to selective outcome reporting (Paxson & Schady, 2010 [28]), all others raised some concerns due to self-reported outcomes, not downgraded because high-risk paper did not contribute to a large extent in overall effect.

e. Moderate, statistically significant, heterogeneity (I2 = 50.11%) and mostly overlapping Cis.

f. Low statistical heterogeneity (I2 = 12.15%), overlapping Cis.

g. High statistical heterogeneity (I2 = 94.03%), CIs not overlapping.

h. No serious threats to directness.

i. CI does not cross line-of-no-effect and is consistent with small positive effect; sample size exceeds 4,000.

j. Analysis underpowered, CI crosses line-of-no-effect.

k. CI crosses line-of-no-effect and is wide enough to cover possibility of meaningful reduction and small increase in stress.

l. Some indication of publication bias, but results are robust to sensitivity analysis.

m. Downgraded since publication bias could not be assessed.

## 4.2) Implications for research and practice

This review contributes to the emergent literature on the links between income increases, poverty alleviation, and mental health. Its findings align with evidence from high-income countries suggesting a causal effect of positive income shocks on mental health [94–97]. Moreover, the small improvements in depression/anxiety symptoms lend some weight to the hypothesis that poverty reduction strengthens psychological health. Hence, intervening on the social causation pathway, by improving the economic circumstances of individuals, can play a role in addressing the poverty-mental health cycle in LMICs [11].

Furthermore, our findings have important implications for cash transfers as a social protection strategy: providing ongoing financial support to people living in poverty not only improves poverty indicators and school attendance, for example, but also meaningfully impacts depression and anxiety outcomes of beneficiaries. However, there is some indication that potential positive effects may not persist after program cessation. Thus, continued financial support may be necessary to allow for longer-term improvements. Importantly, we did not find evidence of adverse effects on depression or anxiety, with almost all studies showing either positive or null-results on average. However, meta-regression indicates that conditionality diminishes the positive effects of cash transfers on mental health. This could introduce trade-

offs between the negative impact observed in this review and the favorable effects of conditionality found on, for instance, school attendance [29] and health-clinic visits [14].

It is important to note, however, that due to the limited number of studies, further research is needed to equip practitioners and policymakers with a better understanding of the role of conditionality for mental health. Such studies could further analyze the potential of soft conditionality in the form of labelling and messaging, which may suffice to achieve desired goals without putting as much strain on participants [98,99].

### 4.3) Limitations and potential biases in the review process

In line with Shea et al. (2017), several steps were undertaken to reduce the risk of bias in this review, such as pre-specifying methods in a published protocol, including MeSh-terms and truncation in the search syntax, adopting a dual-reviewer approach, and searching grey literature websites. Moreover, we used an innovative meta-analytic method to account for statistical dependency between studies and conducted several robustness tests to confirm the accuracy of our estimates.

Nonetheless, the review is subject to a number of limitations. First, although restricting eligibility to randomized trials increases confidence in causal effects, there are important shortcomings of RCT designs. Smaller-scale, controlled experiments have limited generalizability to larger contexts, due to, e.g., unanticipated equilibrium-effects or implementation challenges at scale [100–103]. Moreover, RCTs are typically uninformative with regard to causal mechanisms underlying observed effects, interactions with contextual factors, and differential effects on subgroups, which may not benefit or even be harmed by an intervention [101,104,105]. Although we were able to explore the role of one potential mechanism–conditionality–the limitations of RCTs also apply to this review. As more trials become available, hypotheses about causal mechanisms and subgroup effects, e.g. along the lines of gender and age, should be investigated using meta-regression. Furthermore, additional analysis of quasi-experimental research and longitudinal studies may improve estimates of the effects of CTPs at larger scale.

Second, meta-regression results are limited by low power as well as problems of confounding inherent to this type of analysis [106,107]. UCTs and CCTs may differ in important ways beyond the element of conditionality, introducing the possibility that unknown confounders are responsible for the observed difference. Moreover, treating conditionality as binary variable may neglect more nuanced cases, e.g. when strong messaging in UCTs acts as quasi-conditionality or conditions are not enforced in CCT-programs [29]. This is especially relevant since studies included in this review did not always monitor conditions, one UCT was perceived as conditional, and there was messaging and labelling in UCT trials. Thus, the results should be interpreted as exploratory. Further research may provide more robust evidence on the role of conditionality for mental health.

Third, as expected for a review spanning different countries, programs, and participant groups, heterogeneity was moderate to substantial in two of the conducted meta-analyses. While meta-regression identified conditionality as one significant source of heterogeneity for depression/anxiety outcomes, the limited number of studies did not allow us to include additional covariates in our meta-regression model to further explore variability, particularly for stress outcomes. Nonetheless, we concur with the view that there is value in conducting and reporting meta-analysis, even when heterogeneity is high, as it can highlight inconsistencies and aid narrative synthesis that may otherwise produce misleading results [108].

Fourth, since included studies relied on self-reported outcomes, they potentially overestimate treatment effects when participants reported improvements due to social desirability and/or interviewer effects [109]. Conversely, self-reports may have led to underreporting due

to the stigma attached to mental disorders in LMICs [37]. In light of the difficulties of objectively measuring CMDs, the included trials may be the best currently available evidence on the causal effects of cash transfers on mental health. To mitigate potential biases, future research should adopt strategies to assess and reduce social desirability bias [109], as applied, for instance, by Roy et al. (2019).

Finally, regarding the generalizability of findings beyond included studies, the following limitations should be noted. With the exception of three studies, most trials were conducted in Sub-Saharan Africa, which reduces applicability to other regions and LMICs. Moreover, most studies focused on rural settings and, thus, their findings may not readily apply to urban areas, which are connected to distinct risk/protective factors for mental health [10]. Moreover, extrapolation of research findings typically requires evidence beyond intervention effectiveness, such as insights into mechanisms, implementation, and contextual factors [110–112], the provision of which is beyond the scope of this review.

## 5) Conclusion

Our findings lend weight to the hypothesis that poverty alleviation can play a role in strengthening psychological health of people living in poverty in LMICs. In particular, our analysis shows that providing populations living in poverty with cash transfers leads to improvements of depression and anxiety disorders. However, these benefits may not be sustained once the financial support ends. The findings also indicate that conditionality might diminish the positive mental health effects. Policies aiming to address the poverty-mental health cycle should therefore consider unconditional, longer-term support to populations living in poverty. No firm conclusions can be drawn regarding the effects of CTPs on recipients' stress. Thus, more research is needed to furnish the analysis of stress outcomes, the long-term effects after program cessation and the comparison between conditional and unconditional CTPs.

## Supporting information

**S1 File. S1-S5 Files vrevised.**
(DOCX)

**S2 File. S6 File vrevised.**
(DOC)

**S3 File. S7 File.**
(PDF)

## Author Contributions

**Conceptualization:** Clara Wollburg.

**Formal analysis:** Clara Wollburg.

**Methodology:** Clara Wollburg, Janina Isabel Steinert, Elizabeth Nye.

**Project administration:** Clara Wollburg.

**Supervision:** Aaron Reeves, Elizabeth Nye.

**Validation:** Janina Isabel Steinert.

**Visualization:** Clara Wollburg.

**Writing – original draft:** Clara Wollburg.

**Writing – review & editing:** Janina Isabel Steinert, Aaron Reeves, Elizabeth Nye.

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
