## [Decision Letter · Decision Letter 0]

3 Mar 2021

PONE-D-21-00721

Do cash transfers alleviate common mental disorders in low- and middle-income countries? A systematic review and meta-analysis

PLOS ONE

Dear Dr. Wollburg,

Thank you for submitting your manuscript to PLOS ONE. After careful consideration, we feel that it has merit but does not fully meet PLOS ONE’s publication criteria as it currently stands. Therefore, we invite you to submit a revised version of the manuscript that addresses the points raised during the review process.

Please incorporate recommendations for revisions by reviewer 1 and reviewer 2. Please pay specific attention to the following issues: (1) Depression and anxiety are distinct mental health concepts, why are the authors pooling these concepts? Can the authors share findings where they keep them separate? (2) The authors note that mixed-effects were used in the meta-regression. Why mixed-effects and not random effects as motivated above for the meta-analysis? (3) Can the authors say anything about the amount of cash transferred and how this amount varies between studies? (4) the introduction mentions the aim is to understand the effect of CTPs on depression, anxiety, and stress whereas the research objectives do not state stress. Please correct. (5) the coefficient-estimate for CCTs is indicating on average lower mental health scores than for UCTs however from this estimate one cannot deduct that CCTs worsen mental health outcomes in general. The latter can be identified by having separate analysis, one for CCTs and one for UCTs. (6) A number of relationships are introduced between poverty and mental health. However, this paper does not investigate these relationships as such; instead they are investigating the impact of a particular type of poverty-alleviation instruments (cash transfers) on mental health. Therefore, some explanation is needed on some of the assumption made. A key assumption seems to be that because of the relationship between poverty and mental health, programs that reduce poverty might also reduce mental health problems (there are also many reasons why this might not be the case especially as poverty is a multi-dimensional concept that refer to more than just not having money) (7) Authors state that there is no link between common mental disorders and employment; they cite a review by Lund et al; however the review is much more cautious and explains the lack of consistent evidence is due to measurement challenges; overall, a lot of literature from high income countries suggesting a strong link.

We look forward to receiving your revised manuscript.

Kind regards,

M. Harvey Brenner, PhD

Academic Editor

PLOS ONE

Journal Requirements:

Reviewers' comments:

Reviewer's Responses to Questions

**Comments to the Author**

1. Is the manuscript technically sound, and do the data support the conclusions?

Reviewer #1: Yes

Reviewer #2: Yes

2. Has the statistical analysis been performed appropriately and rigorously? 

Reviewer #1: Yes

Reviewer #2: Yes

3. Have the authors made all data underlying the findings in their manuscript fully available?

Reviewer #1: Yes

Reviewer #2: Yes

4. Is the manuscript presented in an intelligible fashion and written in standard English?

Reviewer #1: Yes

Reviewer #2: Yes

5. Review Comments to the Author

Reviewer #1: Dear authors,

Thank you for writing such an interesting and important research paper. I have enjoyed reading the draft and have the following comments that hopefully help to improve the submission.

Introduction:

[1]“It is therefore critical to shed further light on the potential causal effects of both specific poverty alleviation programs (e.g. cash transfers) and particular dimensions of poverty (e.g. income) on mental disorders in LMICs. We aim to address this gap in the literature by undertaking a systematic review and meta-analysis of the mental health effects of cash transfer programs (CTPs).” The existing studies perform the causal analysis (here subset of RCTs), the review itself assesses whether the findings can be generalised beyond the setting of the study, i.e. using random-effects: to identify the estimated average treatment effect. I would argue this is the contribution of the meta-analysis not to identify causality. Rephrase where applicable.

Section 1.1:

[2] The section dives into pathways of effect, which leads the reader astray, somehow expecting a systematic review of pathways not outcomes of CTPs on CMDs. I suggest being more concise, and to focus on describing firstly from a theoretical point the effect of cash transfers on CMDs, either the social causation hypothesis discussed in various papers by Crick Lund (Lund et al. 2011) or arguing from the Grossman model of health which you can check in your quoted paper (Ohrnberger et al. 2020) would make a good opening paragraph, and then secondly to mention in one paragraph evidence that indicates there may be positive or negative outcomes which you have in good detail in the paper, just boiling it down. I am not sure if figures 1 and 2 add much here.

Lund et al. 2011: https://doi.org/10.1016/S0140-6736(11)60754-X

Ohrnberger et al. 2020: doi: 10.1016/j.socscimed.2020.113181

[3] Consistency: the introduction mentions the aim is to understand the effect of CTPs on depression, anxiety, and stress whereas the research objectives do not state stress. Please correct.

Section 2.1:

[4] Intervention (i): Can the authors say anything about the amount of cash transferred and how this amount varies between studies?

[5] Intervention (ii): More detail on the comparability of various study samples, i.e. what are the definitions of “poor” in the various studies and are concepts of poverty similar across the different studies, giving comparable samples? Are the samples drawn from rural or urban or equally balanced in both settings?

Section 2.3:

[6] Meta-analysis/meta-regression: The authors note that mixed-effects were used in the meta-regression. Why mixed-effects and not random effects as motivated above for the meta-analysis? Further, whilst random effects are correct, I would suggest using meta-regression and re-estimate the meta-analysis controlling for factors related to the interventions such as duration or cash amount, time since programme stopped and other contextual factors such as region/country. This will help to improve the precision of the estimation results and decrease the heterogeneity in the effect, especially important for stress.

Section 3.3:

[7] Depression and anxiety are distinct mental health concepts, why are the authors pooling these concepts? Can the authors share findings where they keep them separate?

Section 3.4:

[8] Lines 534ff: the coefficient-estimate for CCTs is indicating on average lower mental health scores than for UCTs however from this estimate one cannot deduct that CCTs worsen mental health outcomes in general. The latter can be identified by having separate analysis, one for CCTs and one for UCTs.

Section 4.1:

[9] Lines 592-595: I suggest rephrasing the statement; cash transfers are not targeted to mentally ill populations and provide an average treatment effect whereas treatment with anti-depressants are targeted towards a sub-sample. Effects will be larger for those in need which are at the end of the effect distribution. To illustrate, a recent analysis which moved beyond the mean found a four-times increased of a positive cash-transfer programme effect on mental health for those with worst mental health conditions compared to the mean-effect (Ohrnberger et al. 2020).

https://doi.org/10.1093/heapol/czaa079

Section 4.2:

[10] “CTPs can positively impact depression and anxiety, but that effects are small compared to treatment directly targeting mental disorders. Thus, if the aim of policymakers and practitioners is to meaningfully alleviate CMDs, monetary transfers may be complemented with additional support, such as psychological therapy (100. Second, there is some indication that potential positive effects may not persist after program cessation.” The findings need to be interpreted within the context. CTP effects using a random poor population sample are not comparable to directly targeted mental health interventions of samples that are mentally-ill, who are not representing the average poor-population- they may be of the average poor and mentally-ill population. The findings in this study point towards something else. Income-poor individuals exposed to ongoing cash transfer programmes show less symptoms for depression and anxiety disorders. Thusly, a meaningful side-effect of cash transfer programmes is that income-security has an immediate effect on mental health which ceases once the cash-transfer has stopped. This general average treatment effect estimate is useful to understand the wider benefits of CPTs for policymakers and the allocation of scarce resources. To make an even stronger point, the findings should be discussed in light of theoretical foundations (once added in section 1.1)

Section 4.3:

[11] “Furthermore, additional analysis of quasi-experimental research may improve estimates of the effects of CTPs at larger scale.” These studies exist however the review selected only RCTs. This may be added as limitation to the paper.

Conclusion:

[12] Re lines: [726-728]: The analysis found that cash transfers reduce the scores on average however this may not necessarily imply reductions in depression or anxiety disorders. For instance, the CES-D scale has a cut-off point to determine whether an individual is depressed. If the average is driven by reductions far below the threshold disorders are unaffected. I would suggest to rather state that cash transfers improve mental health and anxiety outcomes than reduce the disorders.

Reviewer #2: Overall, this is an important paper, which presents interesting findings on positive as well as potentially adverse effects of cash transfer programmes on mental health, and which has been conducted robustly. The authors draw important conclusions and provide additional value by comparing their results with treatment effects and effects of other relevant outcomes. Whilst the review adds importantly to the literature on the effects of cash programme on mental health, authors do no consider how they findings relate to two recent reviews, which has been published on this topic:

Ridley MW, Rao G, Schilbach F, Patel VH. Poverty, Depression, and Anxiety: Causal Evidence and Mechanisms (No. w27157).National Bureau of Economic Research 2020.https://pubmed.ncbi.nlm.nih.gov/33303583/. (It seems worth highlighting that Ridley et al do not mention potentially adverse effects, which is an important finding of this review.)

McGuire J, Kaiser C, Bach-Mortensen A. The impact of cash transfers on subjective well-being and mental health in low-and middle-income countries: A systematic review and meta-analysis. 2020.

https://www.happierlivesinstitute.org/uploads/1/0/9/9/109970865/cash_transfer_meta-analysis_1.39.pdf

In addition, in my view a number issues would need to be addressed before it can be published:

Abstract: The introduction to the topic currently does not make an explicit distinction between the link between poverty and mental health and the link between poverty alleviation interventions and mental health therefore indirectly assuming that this is the same.

Abstract onwards:

Describing people as poor can be stigmatising, as it defines the person and does not allow a dynamic concept in which people can move out of poverty. Authors might consider using the term people living in poverty rather than poor people or populations

lines 51 onwards

The relationships between poverty and mental health are complex and authors should acknowledge this here - higher risk version and reduced self-control are two mechanisms - others include stigma and discrimination, health expenditure (Lund et al)

lines 54 onwards

A number of relationships are introduced between poverty and mental health. However, this paper does not investigate these relationships as such; instead they are investigating the impact of a particular type of poverty-alleviation instruments (cash transfers) on mental health. Therefore, some explanation is needed on some of the assumption made. A key assumption seems to be that because of the relationship between poverty and mental health, programmes that reduce poverty might also reduce mental health problems (there are also many reasons why this might not be the case especially as poverty is a multi-dimensional concept that refer to more than just not having money)

lines 56

Authors state that there is no link between common mental disorders and employment; they cite a review by Lund et al; however the review is much more cautious and explains the lack of consistent evidence is due to measurement challenges; overall, a lot of literature from high income countries suggesting a strong link.

lines 63 onwards

It seems odd that the author argue that there is not evidence on the impact of cash transfers on mental health but then suggest to conduct a systematic review. Paragraph lines 54 to 67 are not clear. In particular, the evidence gap need to be made clearer, especially considering that two reviews have been published in this area recently. It requires some clear explanation, as to what the current evidence does and does not cover, and how this review seeks to address those.

lines 123 onwards

It would be helpful to understand if this applies because cash amounts are too small or in areas where there is no access to healthier food etc. - especially as this seem at odds with findings from studies that show that subsidies lead to healthier food consumptions as people can afford more fresh food (e.g. An, R. (2013). Effectiveness of subsidies in promoting healthy food purchases and consumption: A review of field experiments. Public Health Nutrition, 16(7), 1215-1228. doi:10.1017/S1368980012004715)

lines 112 onwards

I am not sure how the logic models fit within the scope of the paper, and I would argue they distract rather than contribute to the value of the paper. Especially as the mechanisms are not the the topic of the review and they are compilation by the author without clear methods on how they they were derived, e.g. whether there is agreement from stakeholders.

lines 148 onwards

It is not clear why children were excluded even though they are the main target group of many cash transfer programmes, which specifically seek to develop human capital of children.

lines 176 onwards

The choice of mental heath indicators is not clear. the author refers to the literature on poverty-mental health cycle but not clear how this then led to the choice of mental health indicators. Some clarity here would be good. In particular it is not clear why for example conduct problems or suicide was not not included even though studies exist that demonstrate positive effects.

If authors were only interested in common mental mental health problems this should be made clear in the introduction and in the discussion as conclusions. Some rational would need to be provided with regard to this choice.

lines 184 onwards

Whilst the reason for only including RCTs is stated, it should be noted that this leaves out importance evidence. RCTs are not necessarily the gold standard for evaluating large complex interventions such as cash transfer programmes. In particular, this leaves out evidence from important longitudinal studies that cover very large proportion of people receiving cash transfers.

lines 581 onwards

Considering that authors find that the boundaries between conditional and unconditional programmes are blurred (line 389 onwards), there is a question whether results can be reliably drawn on differences in effects between those two types of programmes. It would be helpful if authors could elaborate on this.

6. PLOS authors have the option to publish the peer review history of their article (what does this mean?). If published, this will include your full peer review and any attached files.

Reviewer #1: No

Reviewer #2: No

---

## [Author Response · Author response to Decision Letter 0]

2 Sep 2021

Dear Reviewers, dear Editors,

We wish to thank you for the invitation to submit a revised version of our systematic review and the valuable comments and detailed feedback, from which our review has greatly benefitted. To address the issues raised, we incorporated suggestions and feedback in a revised manuscript, which we have provided in a marked-up version with track changes as well as an unmarked version. Based on your feedback, our main revisions were:

 1. To provide a stronger theoretical grounding to motivate our research question and interpret our findings

 2. To clarify the research gap our review addresses, especially in relation to the recent reviews by Ridley et al. (2020) and McGuire et al. (2020)

 3. To motivate the use of a mixed-effects meta-regression

 4. To furnish our description of included studies with further information on the amount of cash transferred and how this amount varies between studies

 5. To elaborate why depression and anxiety outcomes were meta-analyzed jointly

Below, we discuss our revisions in detail and provide a point-by-point response to each comment.

 

Point-by-point response to comments made by Reviewer 1

Introduction:

[1] “It is therefore critical to shed further light on the potential causal effects of both specific poverty alleviation programs (e.g. cash transfers) and particular dimensions of poverty (e.g. income) on mental disorders in LMICs. We aim to address this gap in the literature by undertaking a systematic review and meta-analysis of the mental health effects of cash transfer programs (CTPs).” The existing studies perform the causal analysis (here subset of RCTs), the review itself assesses whether the findings can be generalised beyond the setting of the study, i.e. using random-effects: to identify the estimated average treatment effect. I would argue this is the contribution of the meta-analysis not to identify causality. Rephrase where applicable.

We absolutely agree that although systematic reviews and meta-analyses can increase confidence in causal effects of interventions (Greenhalgh, 1997), they do not themselves constitute a form of causal analysis. We have rephrased accordingly, e.g., “we collate the causal evidence” (Section 1, line 69) and “we synthesize the evidence from randomized controlled trials” (Section 1, line 97).

Section 1.1:

[2] The section dives into pathways of effect, which leads the reader astray, somehow expecting a systematic review of pathways not outcomes of CTPs on CMDs. I suggest being more concise, and to focus on describing firstly from a theoretical point the effect of cash transfers on CMDs, either the social causation hypothesis discussed in various papers by Crick Lund (Lund et al. 2011) or arguing from the Grossman model of health which you can check in your quoted paper (Ohrnberger et al. 2020) would make a good opening paragraph, and then secondly to mention in one paragraph evidence that indicates there may be positive or negative outcomes which you have in good detail in the paper, just boiling it down. I am not sure if figures 1 and 2 add much here.

Lund et al. 2011: https://doi.org/10.1016/S0140-6736(11)60754-X

Ohrnberger et al. 2020: doi: 10.1016/j.socscimed.2020.113181

Thank you for this valuable feedback and excellent suggestions. Following your suggestion, we have rewritten our introduction and incorporated the social causation hypothesis as a theoretical foundation to motivate our research question:

“The association between poverty and mental health is not a spurious correlation, however, and several studies indicate that economic poverty causes and exacerbates psychological disorders (8,12). Together, these studies support the social causation hypothesis, which states that living in conditions of poverty is connected to a multitude of severe stressors that increase vulnerability to mental disorders (6,10). Accordingly, interventions that reduce monetary poverty may intervene on the social causation pathways and, thus, can help alleviate the burden of common mental disorders in targeted populations. 

In this systematic review, therefore, we collate the causal evidence on the psychological impacts of one such intervention – the provision of cash transfers – to investigate whether monetary poverty alleviation can play a role in strengthening psychological health in LMICs.” (Section 1, lines 62 – 71)

Moreover, we removed Figures 1 and 2 and condensed the information provided on pathways:

“Cash transfer programs (CTPs), which provide a financial safety net to people in poverty, are hypothesized to improve mental health outcomes by, e.g., reducing financial strain and instability, which has been linked to heightened stress and depressive symptoms in LMICs (3). Cash transfers were also shown to facilitate social inclusion (4,13) and to improve physical health indicators (14–16), factors which have been linked to stress, anxiety and depressive symptoms in LMICs (3,6,17).

On the other hand, CTPs may adversely affect mental health, as they can aggravate social exclusion (18) and are sometimes associated with increased health risks, e.g. higher body mass index (19,20), thereby potentially increasing risks of mental ill-health. Finally, making CTPs conditional on certain behaviors can increase feelings of stress when conditions are overly demanding (21). Conditionality may further exclude particularly vulnerable individuals, which could increase mental health inequalities (22,23).” (Section 1, lines 83 – 94) 

[3] Consistency: the introduction mentions the aim is to understand the effect of CTPs on depression, anxiety, and stress whereas the research objectives do not state stress. Please correct.

Kindly refer to line 101, where we stated as research objective 2: 

“2. To examine how CTPs impact recipients’ stress, compared to inactive control groups.” (Section 1, line 101)

Section 2.1:

[4] Intervention (i): Can the authors say anything about the amount of cash transferred and how this amount varies between studies?

This is an important point, thank you. We have provided additional information on the transferred cash amount in relative terms (expenditure) as well as absolute terms (cash amount in USD) and included the cash amount for each cash transfer program as part of Supplementary Appendix S3.

“Transfer values varied between US$144 and US$1,525 over the length of the program, which is equivalent to approximately 9%–23% of participants monthly consumption (calculations based on 15 studies for which data was available, see Supplementary Appendix S3 for cash values per program)” (Section 3.1.1, lines 561 – 572)

[5] Intervention (ii): More detail on the comparability of various study samples, i.e. what are the definitions of “poor” in the various studies and are concepts of poverty similar across the different studies, giving comparable samples? Are the samples drawn from rural or urban or equally balanced in both settings?

Thank you very much for these comments regarding the comparability of study samples. We discuss the distribution of urban vs rural settings in lines 546 – 548:

“With the exception of the Urban Microinsurance project (72), all trials were either conducted in rural areas (13 of 17 studies) or focused on both urban and rural households (3 of 17 studies)” (Section 3.1.1, lines 546 – 548)

Accordingly, we have noted as limitation of our review:

“Moreover, most studies focused on rural settings and, thus, their findings may not readily apply to urban areas, which are connected to distinct risk/protective factors for mental health (10).” (Section 4.3, lines 993 – 995) 

Regarding the definitions of poverty employed in included studies, we have added the following paragraph:

“Transfers were targeted to low-income and/or deprived communities, as indicated by, e.g., low monthly household expenditure and consumptions (33,74,75,77–79), inability to meet basic needs (32,73), food insecurity (74,80), low educational attainment and high HIV risk (21,71).” (Section 3.1.1, lines 580 – 583)

Section 2.3:

[6] Meta-analysis/meta-regression: The authors note that mixed-effects were used in the meta-regression. Why mixed-effects and not random effects as motivated above for the meta-analysis? Further, whilst random effects are correct, I would suggest using meta-regression and re-estimate the meta-analysis controlling for factors related to the interventions such as duration or cash amount, time since programme stopped and other contextual factors such as region/country. This will help to improve the precision of the estimation results and decrease the heterogeneity in the effect, especially important for stress.

Thank you for these helpful suggestions. We have carefully considered whether to include additional covariates, such as cash amount or length of exposure, in our meta-regression model. However, due to the limited number of available studies, this would result in low power and would thus likely reduce the accuracy of our estimate. We therefore decided not to include further covariates. Accordingly, we have emphasized this point in the limitations section:

“Third, as expected for a review spanning different countries, programs, and participant groups, heterogeneity was moderate to substantial in two of the conducted meta-analyses. While meta-regression identified conditionality as one significant source of heterogeneity for depression/anxiety outcomes, the limited number of studies did not allow us to include additional covariates in our meta-regression model to further explore variability, particularly for stress outcomes. Nonetheless, we concur with the view that there is value in conducting and reporting meta-analysis, even when heterogeneity is high, as it can highlight inconsistencies and aid narrative synthesis that may otherwise produce misleading results (107).” (Section 4.3, lines 969 – 976)

Regarding your first comment: We follow Tanner-Smith, Tipton, & Polanin (2016) in using the term mixed-effects model for our RVE meta-regression. However, in practice the terms mixed-effects and random-effects meta-regression are interchangeable (see for instance the use of the terms in Stanley & Doucouliagos (2017)). As opposed to fixed-effects meta-regression, the mixed- effects model allows us to accommodate both within-study variation and between-study variation. We have clarified our rationale for choosing a mixed-effects model in lines 451 – 460:

“To explore the relative effects of conditional vs unconditional CTPs, we conducted RVE meta-regression with a mixed-effects model so as to accommodate within-study and between-study variation. The meta-regression model can be written as:

y_(ij )= β_0+ β_1 (〖conditionality)〗_ij+ u_j+ e_ij

Where the covariate β_1, which represents a dummy variable specifying whether the CTP was conditional, is added to the RVE-model in above section (55). Meta-regression was performed for depression and anxiety only, since all studies analyzing stress outcomes examined unconditional CTPs.” (Section 2.3, lines 451 – 460)

Section 3.3:

[7] Depression and anxiety are distinct mental health concepts, why are the authors pooling these concepts? Can the authors share findings where they keep them separate?

Thank you for this important point. We have debated whether to analyze depression and anxiety separately. However, as explained in section 3.3, the questionnaires used for depression and anxiety, particularly the GHQ-12, PHQ-9 and CES-D and have considerable overlap. For instance, GHQ-12, CES-D and PHQ-9 all include items on sleep disturbances, feelings of depression, self-worth, or ability to concentrate (Goldberg et al., 1997; Kroenke & Spitzer, 2002; Radloff, 1977). Separating anxiety and depression outcomes would have been further complicated by the fact that the GHQ-12 is a composite measure for common mental disorders, detecting both depressive and anxiety symptoms. We therefore concluded that the questionnaires measured overlapping constructs and, thus, they were meta-analyzed jointly.

For reference see section 3.3., lines 646 – 657:

“11 studies reported depression outcomes, using CES-D, PHQ-9 and CDI-questionnaires, one study measured anxiety using the CMAS-questionnaire, and two studies measured common mental disorders, including depressive/anxiety symptoms, using the GHQ-12. Since CES-D, PHQ, and GHQ-measures have considerable overlap and exhibit similar response patters (86), it was concluded that anxiety and depression represent overlapping constructs. For instance, GHQ-12, CES-D and PHQ-9 all include items on sleep disturbances, feelings of depression, self-worth, or ability to concentrate (87–89). Thus, anxiety and depression outcomes were meta-analyzed jointly.” (Section 3.3., lines 646 – 657)

Section 3.4:

[8] Lines 534ff: the coefficient-estimate for CCTs is indicating on average lower mental health scores than for UCTs however from this estimate one cannot deduct that CCTs worsen mental health outcomes in general. The latter can be identified by having separate analysis, one for CCTs and one for UCTs.

Thank you for this important feedback – we have rephrased accordingly, e.g.:

“The CCT-estimate suggests that adding conditionality may decrease positive effects on depression and anxiety (dpooled=0.10; 95%-CI: 0.07, 0.13; p<0.01)” (Section 3.4, lines 740 – 741)

“The findings also indicate that conditionality might diminish the positive mental health effects.” (Section 5, lines 1006 – 1007)

Section 4.1:

[9] Lines 592-595: I suggest rephrasing the statement; cash transfers are not targeted to mentally ill populations and provide an average treatment effect whereas treatment with anti-depressants are targeted towards a sub-sample. Effects will be larger for those in need which are at the end of the effect distribution. To illustrate, a recent analysis which moved beyond the mean found a four-times increased of a positive cash-transfer programme effect on mental health for those with worst mental health conditions compared to the mean-effect (Ohrnberger et al. 2020).

https://doi.org/10.1093/heapol/czaa079

Many thanks for this helpful comment and the suggested citation. Although contextualizing effect sizes can aid interpretation, we absolutely agree that there is limited comparability of samples from CTP studies and samples from mentally-ill populations in LMICs. In response to your comment, we therefore decided to remove the reference to the systematic reviews by Moncrieff et al. (2004) and Purgato et al. (2018).

Section 4.2:

[10] “CTPs can positively impact depression and anxiety, but that effects are small compared to treatment directly targeting mental disorders. Thus, if the aim of policymakers and practitioners is to meaningfully alleviate CMDs, monetary transfers may be complemented with additional support, such as psychological therapy (100. Second, there is some indication that potential positive effects may not persist after program cessation.” The findings need to be interpreted within the context. CTP effects using a random poor population sample are not comparable to directly targeted mental health interventions of samples that are mentally-ill, who are not representing the average poor-population- they may be of the average poor and mentally-ill population. The findings in this study point towards something else. Income-poor individuals exposed to ongoing cash transfer programmes show less symptoms for depression and anxiety disorders. Thusly, a meaningful side-effect of cash transfer programmes is that income-security has an immediate effect on mental health which ceases once the cash-transfer has stopped. This general average treatment effect estimate is useful to understand the wider benefits of CPTs for policymakers and the allocation of scarce resources. To make an even stronger point, the findings should be discussed in light of theoretical foundations (once added in section 1.1)

Many thanks for these valuable points. In line with these suggestions and your comments [1] and [2], we put a greater emphasis on the theoretical foundations of our research question and revised the introduction and discussion section accordingly. This is reflected most notably in section 4.2, which we rewrote as follows: 

“This review contributes to the emergent literature on the links between income increases, poverty alleviation, and mental health. Its findings align with evidence from high-income countries suggesting a causal effect of positive income shocks on mental health (93–96). Moreover, the small improvements in depression/anxiety symptoms lend some weight to the hypothesis that poverty reduction strengthens psychological health. Hence, intervening on the social causation pathway, by improving the economic circumstances of individuals, can play a role in addressing the poverty-mental health cycle in LMICs (11).

Furthermore, our findings have important implications for cash transfers as a social protection strategy: providing ongoing financial support to people living in poverty not only improves poverty indicators and school attendance, for example, but also meaningfully impacts depression and anxiety outcomes of beneficiaries. However, there is some indication that potential positive effects may not persist after program cessation. Thus, continued financial support may be necessary to allow for longer-term improvements. Importantly, we did not find evidence of adverse effects on depression or anxiety, with almost all studies showing either positive or null-results on average. However, meta-regression indicates that conditionality diminishes the positive effects of cash transfers on mental health. This could introduce trade-offs between the negative impact observed in this review and the favorable effects of conditionality found on, for instance, school attendance (29) and health-clinic visits (14).” (Section 4.2, lines 856 - 868)

Section 4.3:

[11] “Furthermore, additional analysis of quasi-experimental research may improve estimates of the effects of CTPs at larger scale.” These studies exist however the review selected only RCTs. This may be added as limitation to the paper.

Agreed. We have reflected on the limitations of RCT evidence in the limitations section 4.3:

“Nonetheless, the review is subject to a number of limitations. First, although restricting eligibility to randomized trials increases confidence in causal effects, there are important shortcomings of RCT designs. Smaller-scale, controlled experiments have limited generalizability to larger contexts, due to, e.g., unanticipated equilibrium-effects or implementation challenges at scale (100–102). Moreover, RCTs are typically uninformative with regard to causal mechanisms underlying observed effects, interactions with contextual factors, and differential effects on subgroups, which may not benefit or even be harmed by an intervention (100,103,104). Although we were able to explore the role of one potential mechanism – conditionality – the limitations of RCTs also apply to this review. As more trials become available, hypotheses about causal mechanisms and subgroup effects, e.g. along the lines of gender and age, should be investigated using meta-regression. Furthermore, additional analysis of quasi-experimental research and longitudinal studies may improve estimates of the effects of CTPs at larger scale.” (Section 4.3, lines 945 – 956)

Conclusion:

[12] Re lines: [726-728]: The analysis found that cash transfers reduce the scores on average however this may not necessarily imply reductions in depression or anxiety disorders. For instance, the CES-D scale has a cut-off point to determine whether an individual is depressed. If the average is driven by reductions far below the threshold disorders are unaffected. I would suggest to rather state that cash transfers improve mental health and anxiety outcomes than reduce the disorders.

Yes, we agree and have reworded as suggested. Thank you for this helpful suggestion!

Point-by-point response to comments made by Reviewer 2

[1] Overall, this is an important paper, which presents interesting findings on positive as well as potentially adverse effects of cash transfer programmes on mental health, and which has been conducted robustly. The authors draw important conclusions and provide additional value by comparing their results with treatment effects and effects of other relevant outcomes. Whilst the review adds importantly to the literature on the effects of cash programme on mental health, authors do no consider how they findings relate to two recent reviews, which has been published on this topic:

Ridley MW, Rao G, Schilbach F, Patel VH. Poverty, Depression, and Anxiety: Causal Evidence and Mechanisms (No. w27157). National Bureau of Economic Research 2020. https://pubmed.ncbi.nlm.nih.gov/33303583/. (It seems worth highlighting that Ridley et al do not mention potentially adverse effects, which is an important finding of this review.)

McGuire J, Kaiser C, Bach-Mortensen A. The impact of cash transfers on subjective well-being and mental health in low-and middle-income countries: A systematic review and meta-analysis. 2020.

https://www.happierlivesinstitute.org/uploads/1/0/9/9/109970865/cash_transfer_meta-analysis_1.39.pdf

Many thanks for this valuable feedback and the provided references. To clarify how our review relates to that of Ridely et al. (2020) and McGuire et al. (2020), we have rewritten the introduction and added the following paragraphs:

“To our knowledge, this is the most comprehensive systematic review and meta-analysis on the mental health effects of cash transfers. Three earlier reviews considered studies on cash transfers and common mental disorders (11,14,24) and found some encouraging effects on children’s cortisol levels and aggressive symptoms, and positive, but insignificant, effects on depression and anxiety of children and adults (25–28). Yet, since relevant studies published over the last decade are not included, these reviews do not provide a timely, comprehensive overview of current evidence. Moreover, none of the earlier reviews conducted a meta-analysis to offer a quantitative summary of the results because of the limited number of available studies. Two recent papers have synthesized more current evidence. Ridley et al. (2020) conducted a literature review and meta-analysis of the psychological impacts of anti-poverty programs, including cash transfers (12). Our review extends their work by using a systematic review methodology and comprehensive search strategy, which allowed us to retrieve additional relevant studies. Moreover, our review pushes the literature forward by providing a comparison of the impacts of conditional and unconditional cash transfers on mental health. Making the receipt of transfers conditional on requirements, such as regular school attendance of children and health-clinic visits, has been shown to increase effectiveness for those outcomes (14,29). However, adding an element of conditionality may also lead to unintended consequences, such as increased stress among recipients (21) or adverse effects for those loosing eligibility (22). As a result, conditionality might negatively affect psychological health and could exacerbate mental health inequalities. Understanding the psychological effects of conditionality has therefore important implications for CTPs as a social protection strategy. 

Secondly, McGuire et al. (2020) conducted a systematic review of quasi-experimental and experimental studies of CTPs and their effects on subjective wellbeing and mental health (30). Our review complements their work in two ways. Firstly, we were able to retrieve additional grey literature papers (31–33). In light of the likelihood of publication bias detected in this review (see section 3.6), this is an important addition and allows us to paint a more comprehensive picture of the effects. Moreover, we use an advanced meta-analytic method – robust variance estimation (RVE) – which corrects for dependency between effect sizes, thus producing more accurate estimations of the overall effect than other meta-analytic models (34). Moreover, RVE allowed us to integrate all reported effect sizes, thus avoiding loss of information. This is highly relevant for the cash transfer literature as studies often analyze data from the same CTP or report multiple effect sizes per sample.” (Section 1, lines 248 – 279)

Abstract

[2] The introduction to the topic currently does not make an explicit distinction between the link between poverty and mental health and the link between poverty alleviation interventions and mental health therefore indirectly assuming that this is the same.

Thanks a lot for this point. We agree and have rewritten the Abstract to make the distinction between the poverty-mental health link vs. the poverty intervention-mental health link clearer:

“A large literature has demonstrated the link between poverty and mental ill-health. Yet, the potential causal effects of poverty alleviation measures on mental disorders are not well-understood. In this systematic review, we summarize the evidence of the effects of a particular kind of poverty alleviation mechanism on mental health: the provision of cash transfers in low- and middle-income countries. (…)” (Abstract, lines 15 – 19)

Abstract onwards:

[3] Describing people as poor can be stigmatising, as it defines the person and does not allow a dynamic concept in which people can move out of poverty. Authors might consider using the term people living in poverty rather than poor people or populations

Thank you for this important feedback – we now use the term “people living in poverty” throughout the review.

Lines 51 onwards

[4] The relationships between poverty and mental health are complex and authors should acknowledge this here - higher risk version and reduced self-control are two mechanisms - others include stigma and discrimination, health expenditure (Lund et al)

Many thanks for this suggestion. In order to account for the multiple mechanisms underlying the mental health-poverty-relationship, we have reworded the paragraph as follows:

“The high prevalence of mental disorders is not only damaging in itself, but can contribute to the persistence of poverty through, e.g., lower productivity, stigmatization, higher health expenditures, and reduced self-control (8,10,11).“ (Section 1, lines 58 – 61)

Lines 54 onwards

[5] A number of relationships are introduced between poverty and mental health. However, this paper does not investigate these relationships as such; instead they are investigating the impact of a particular type of poverty-alleviation instruments (cash transfers) on mental health. Therefore, some explanation is needed on some of the assumption made. A key assumption seems to be that because of the relationship between poverty and mental health, programmes that reduce poverty might also reduce mental health problems (there are also many reasons why this might not be the case especially as poverty is a multi-dimensional concept that refer to more than just not having money)

Thank you for raising this point. In response to your feedback, as well as feedback by reviewer 1, we have rewritten the introduction with a reference to the social causation hypothesis, which provides a theoretical grounding for our research question and focuses the introduction more clearly on the poverty intervention-mental health relationship.

“Common mental disorders are a critical public health issue, with more than 250 million people affected by anxiety and over 300 million people suffering from depression globally (1). More than 80% of the global burden of common mental disorders, i.e. depression and anxiety disorders, falls on low- and middle-income countries (LMICs) (1). People who live in poverty in LMICs are disproportionately affected by mental ill-health due to their exposure to risk factors, such as food insecurity, social exclusion, trauma and violence (2–8). Poverty is further correlated with higher levels of stress and lower quality of life and psychosocial wellbeing (8,9). The high prevalence of mental disorders is not only damaging in itself, but can contribute to the persistence of poverty through, e.g., lower productivity, stigmatization, higher health expenditures, and reduced self-control (8,10,11). 

The association between poverty and mental health is not a spurious correlation, however, and several studies indicate that economic poverty causes and exacerbates psychological disorders (8,12). Together, these studies support the social causation hypothesis, which states that living in conditions of poverty is connected to a multitude of severe stressors that increase vulnerability to mental disorders (6,10). Accordingly, interventions that reduce monetary poverty may intervene on the social causation pathways and, thus, can help alleviate the burden of common mental disorders in targeted populations. 

In this systematic review, therefore, we collate the causal evidence on the psychological impacts of one such intervention – the provision of cash transfers – to investigate whether monetary poverty alleviation can play a role in strengthening psychological health in LMICs.” (Section 1, lines 52 – 71)

Lines 56

[6] Authors state that there is no link between common mental disorders and employment; they cite a review by Lund et al; however the review is much more cautious and explains the lack of consistent evidence is due to measurement challenges; overall, a lot of literature from high income countries suggesting a strong link.

Many thanks for pointing this out. Due to our rewriting of the introduction section in line with your feedback [5], we no longer discuss the relationship between common mental disorders and employment.

Lines 63 onwards

[7] It seems odd that the author argue that there is not evidence on the impact of cash transfers on mental health but then suggest to conduct a systematic review. Paragraph lines 54 to 67 are not clear. In particular, the evidence gap need to be made clearer, especially considering that two reviews have been published in this area recently. It requires some clear explanation, as to what the current evidence does and does not cover, and how this review seeks to address those.

Thank you for this helpful suggestion. As described in our response to your first comment, we have rewritten the introduction section so as to emphasize the evidence gap our review addresses, particularly in light of the reviews by of Ridely et al. and McGuire et al. Please refer to our response to comment [1] and lines 248 – 279 in the introduction of the revised manuscript. 

Lines 123 onwards

[7] It would be helpful to understand if this applies because cash amounts are too small or in areas where there is no access to healthier food etc. - especially as this seem at odds with findings from studies that show that subsidies lead to healthier food consumptions as people can afford more fresh food (e.g. An, R. (2013). Effectiveness of subsidies in promoting healthy food purchases and consumption: A review of field experiments. Public Health Nutrition, 16(7), 1215-1228. doi:10.1017/S1368980012004715)

Thank you for this interesting point and the provided reference. The referenced studies by Fernald et al. (2008) and Forde et al. (2012), which find an increased obesity risk and higher BMI for cash transfer recipients, do not test the specific mechanisms underlying this association. In order to account for this, we have used a more careful wording and rephrased the sentence in the following way:

“On the other hand, CTPs may adversely affect mental health, as they can aggravate social exclusion (18) and are sometimes associated with increased health risks, e.g. higher body mass index (19,20), thereby potentially increasing risks of mental ill-health.” (Section 1, lines 89 – 91)

Lines 112 onwards

[8] I am not sure how the logic models fit within the scope of the paper, and I would argue they distract rather than contribute to the value of the paper. Especially as the mechanisms are not the the topic of the review and they are compilation by the author without clear methods on how they they were derived, e.g. whether there is agreement from stakeholders.

Many thanks for this valuable feedback. In response to this, and feedback by reviewer 1, we have shortened the paragraph on mechanisms and removed the logic model. Kindly refer to our response to comment [1] by reviewer 1. 

Lines 148 onwards

[9] It is not clear why children were excluded even though they are the main target group of many cash transfer programmes, which specifically seek to develop human capital of children.

Many thanks for this feedback regarding the target population of our review. You rightly note that many cash transfer programs, especially in Latin American countries, are aimed to improve human capital outcomes of children. However, we have chosen to focus our review on adolescents and adults as we wanted to compare the direct effects on those who are recipients of transfers rather than the indirect effects on family members, e.g., children. In response to you feedback, we made this reasoning more explicit in lines 296 – 299:

“This review focuses on people living in poverty, who are recipients of CTPs. Since mental health disorders persisting into adulthood tend to manifest in the ages 12–24 (36), both adults and adolescents, who are CTP recipients, are included. The indirect effects of CTPs on family members of recipients, e.g. children, are not the focus of this review.” (Section 2.1, lines 296 – 299)

Lines 176 onwards

[10] The choice of mental heath indicators is not clear. the author refers to the literature on poverty-mental health cycle but not clear how this then led to the choice of mental health indicators. Some clarity here would be good. In particular it is not clear why for example conduct problems or suicide was not not included even though studies exist that demonstrate positive effects. If authors were only interested in common mental mental health problems this should be made clear in the introduction and in the discussion as conclusions. Some rational would need to be provided with regard to this choice.

Thank you for pointing us to a lack of clarity regarding our choice of outcome indicators. Our review focuses on common mental disorders, i.e. depression and anxiety disorders, as well as stress. In order to emphasize the focus on common mental disorders, we have rephrased the section “outcomes” as follows:

“Our review focuses on common mental disorders, i.e. depressive and anxiety disorders (1), as well as stress. We focus on common mental disorders, as opposed to severe mental illnesses, such as schizophrenia, since socio-economic risk factors are of limited importance and, thus, CPTs may not readily impact more severe mental disorders (11,37). Studies using validated surveys were included in this review, using e.g. the Center for Epidemiologic Studies Depression Scale, the General Health Questionnaire, or Cohen’s perceived stress scale.” (Section 2, lines 319 – 324)

Moreover, we also used the term common mental disorders in the title, stated the outcomes depression, anxiety and stress in the research objectives (see line 95) and put a greater emphasis on common mental disorders in the introduction and discussion, e.g.:

“Common mental disorders are a critical public health issue, with more than 250 million people affected by anxiety and over 300 million people suffering from depression globally (1).” (Section 1, lines 52 – 53)

And lines 787 – 789:

“Can cash transfers play a role in alleviating common mental disorders in LMICs? This systematic review summarized the evidence on the effects of CTPs on stress, anxiety, and depressive symptoms in adults and adolescents who live in poverty.” (Section 4.1, lines 787 – 789)

Lines 184 onwards

[11] Whilst the reason for only including RCTs is stated, it should be noted that this leaves out importance evidence. RCTs are not necessarily the gold standard for evaluating large complex interventions such as cash transfer programmes. In particular, this leaves out evidence from important longitudinal studies that cover very large proportion of people receiving cash transfers.

We absolutely agree that there are important shortcomings to RCTs, especially for evaluating large complex interventions. We have emphasized this point in the limitation section:

“Nonetheless, the review is subject to a number of limitations. First, although restricting eligibility to randomized trials increases confidence in causal effects, there are important shortcomings of RCT designs. Smaller-scale, controlled experiments have limited generalizability to larger contexts, due to, e.g., unanticipated equilibrium-effects or implementation challenges at scale (100–102). Moreover, RCTs are typically uninformative with regard to causal mechanisms underlying observed effects, interactions with contextual factors, and differential effects on subgroups, which may not benefit or even be harmed by an intervention (100,103,104). Although we were able to explore the role of one potential mechanism – conditionality – the limitations of RCTs also apply to this review. As more trials become available, hypotheses about causal mechanisms and subgroup effects, e.g. along the lines of gender and age, should be investigated using meta-regression. Furthermore, additional analysis of quasi-experimental research and longitudinal studies may improve estimates of the effects of CTPs at larger scale.” (Section 4.3, lines 945 – 956)

Lines 581 onwards

[12] Considering that authors find that the boundaries between conditional and unconditional programmes are blurred (line 389 onwards), there is a question whether results can be reliably drawn on differences in effects between those two types of programmes. It would be helpful if authors could elaborate on this.

Yes, this is an important point. We have elaborated on this issue in the limitation section of our review: 

“Moreover, treating conditionality as binary variable may neglect more nuanced cases, e.g. when strong messaging in UCTs acts as quasi-conditionality or conditions are not enforced in CCT-programs (29). This is especially relevant since studies included in this review did not always monitor conditions, one UCT was perceived as conditional, and there was messaging and labelling in UCT trials. Thus, the results should be interpreted as exploratory. Further research may provide more robust evidence on the role of conditionality for mental health.” (Section 4.3, lines 961 – 965)

---

## [Decision Letter · Decision Letter 1]

5 Oct 2021

PONE-D-21-00721R1Do cash transfers alleviate common mental disorders in low- and middle-income countries? A systematic review and meta-analysisPLOS ONE

Dear Clara Wollburg,

Thank you for submitting your manuscript to PLOS ONE. After careful consideration, we feel that it has merit but does not fully meet PLOS ONE’s publication criteria as it currently stands. Therefore, we invite you to submit a revised version of the manuscript that addresses the points raised during the review process. Please address the issues addressed by reviewer number 2.

We look forward to receiving your revised manuscript.

Kind regards,

M. Harvey Brenner, PhD

Academic Editor

PLOS ONE

Journal Requirements:

Reviewers' comments:

Reviewer's Responses to Questions

**Comments to the Author**

1. If the authors have adequately addressed your comments raised in a previous round of review and you feel that this manuscript is now acceptable for publication, you may indicate that here to bypass the “Comments to the Author” section, enter your conflict of interest statement in the “Confidential to Editor” section, and submit your "Accept" recommendation.

Reviewer #2: All comments have been addressed

2. Is the manuscript technically sound, and do the data support the conclusions?

Reviewer #2: Yes

3. Has the statistical analysis been performed appropriately and rigorously? 

Reviewer #2: Yes

4. Have the authors made all data underlying the findings in their manuscript fully available?

Reviewer #2: Yes

5. Is the manuscript presented in an intelligible fashion and written in standard English?

Reviewer #2: Yes

6. Review Comments to the Author

Reviewer #2: I thank the authors for addressing all comments thoroughly. I believe this version of the manuscript is much improved and will be a very valuable piece of work in the field.

This is just to let you know that one additional systematic review and meta-analysis has been published on this topic since the last review and it might be worth adding a sentence or two perhaps in the discussion section:

Zimmerman A, Garman E, Avendano-Pabon M, et al. The impact of cash transfers on mental health in children and young people in low-income and middle-income countries: a systematic review and meta-analysis. BMJ Global Health. 2021 Apr;6(4). DOI: 10.1136/bmjgh-2020-004661. PMID: 33906845; PMCID: PMC80882; https://europepmc.org/article/PMC/PMC8088245

A minor observation is that it would have been extremely helpful to compare the effect sizes with the one of (universal) mental health intervention, and make the paper even more relevant.

7. PLOS authors have the option to publish the peer review history of their article (what does this mean?). If published, this will include your full peer review and any attached files.

Reviewer #2: **Yes: **Annette Bauer

---

## [Author Response · Author response to Decision Letter 1]

24 Oct 2021

Dear Reviewer, 

Thank you again for your valuable feedback and for pointing us to the recent systematic review by Zimmerman et al. on the mental health effects of cash transfers in young people and children. We have incorporated the review as suggested and have added the following paragraph to the introduction:

“Zimmerman et al. (2021) published a systematic review and meta-analysis on the effects of cash transfers on mental health outcomes of children and young people (35). Since Zimmerman and colleagues focus on children and adolescents, however, their analysis does not include many of the studies analyzed in this review.” (Section 1, lines 108 – 111, revised manuscript with track changes)

Moreover, we have included the findings in the discussion section:

“Can cash transfers play a role in alleviating common mental disorders in LMICs? This systematic review summarized the evidence on the effects of CTPs on stress, anxiety, and depressive symptoms in adults and adolescents who live in poverty. Meta-analysis of 11 studies and pooling data from 22,488 participants shows that cash grants significantly improve depression and anxiety post-intervention. Following GRADE-criteria, the overall confidence is judged to be moderate, due to some indication of publication bias and since meta-analyzed trials carried intermediate to high risk of bias. The magnitude of the effect is comparable to meta-analysis on other distal outcomes in the cash transfer literature, e.g. on child labor (91) and test scores (29). Moreover, our effect sizes are similar to those found by Ridley et al. (12) and McGuire et al. (30), who also report significant positive impacts on common mental disorders. Zimmerman et al. (35), on the other hand, find positive, but much smaller and insignificant aggregate effects on depressive symptoms in children and young people. The authors note, however, that the validity of the aggregate effect size is limited due to substantial heterogeneity.” (Section 4.1, lines 579 – 591, revised manuscript with track changes)

You also noted the following: “A minor observation is that it would have been extremely helpful to compare the effect sizes with the one of (universal) mental health intervention, and make the paper even more relevant.”

Many thanks for this suggestion. In our original submission, we did compare effect sizes of our review to reviews looking at the effectiveness of psychological therapies and antidepressants in LMICs (Moncrieff, Wessely, & Hardy, 2004; Purgato et al., 2018). However, reviewer 1 rightly pointed us to the limited comparability of samples from CTP studies and samples from mentally-ill populations in LMICs: “cash transfers are not targeted to mentally ill populations and provide an average treatment effect whereas treatment with anti-depressants are targeted towards a sub-sample. Effects will be larger for those in need which are at the end of the effect distribution. To illustrate, a recent analysis which moved beyond the mean found a four-times increased of a positive cash-transfer programme effect on mental health for those with worst mental health conditions compared to the mean-effect (Ohrnberger et al. 2020). – https://doi.org/10.1093/heapol/czaa079)”

In response to this comment, we decided not to compare effect sizes to those of mental health interventions.

---

## [Decision Letter · Decision Letter 2]

20 Jan 2023

Do cash transfers alleviate common mental disorders in low- and middle-income countries? A systematic review and meta-analysis

PONE-D-21-00721R2

Dear Clara Wollburg,

We’re pleased to inform you that your manuscript has been judged scientifically suitable for publication and will be formally accepted for publication once it meets all outstanding technical requirements.

Kind regards,

M. Harvey Brenner, PhD

Academic Editor

PLOS ONE

Additional Editor Comments (optional):

Reviewers' comments:

Reviewer's Responses to Questions

**Comments to the Author**

1. If the authors have adequately addressed your comments raised in a previous round of review and you feel that this manuscript is now acceptable for publication, you may indicate that here to bypass the “Comments to the Author” section, enter your conflict of interest statement in the “Confidential to Editor” section, and submit your "Accept" recommendation.

Reviewer #2: All comments have been addressed

2. Is the manuscript technically sound, and do the data support the conclusions?

Reviewer #2: Yes

3. Has the statistical analysis been performed appropriately and rigorously? 

Reviewer #2: Yes

4. Have the authors made all data underlying the findings in their manuscript fully available?

Reviewer #2: Yes

5. Is the manuscript presented in an intelligible fashion and written in standard English?

Reviewer #2: Yes

6. Review Comments to the Author

Reviewer #2: All comments have been addressed. Authors have revised their discussion section to refer in detail how their findings relate to recent systematic reviews and meta-analysis, and how this might be explained by methodological differences in scope.

7. PLOS authors have the option to publish the peer review history of their article (what does this mean?). If published, this will include your full peer review and any attached files.

Reviewer #2: **Yes: **Annette Bauer

---

## [Editor Report · Acceptance letter]

27 Jan 2023

PONE-D-21-00721R2 

Do cash transfers alleviate common mental disorders in low- and middle-income countries? A systematic review and meta-analysis 

Dear Dr. Wollburg:

I'm pleased to inform you that your manuscript has been deemed suitable for publication in PLOS ONE. Congratulations! Your manuscript is now with our production department. 

Kind regards, 

on behalf of

Professor M. Harvey Brenner 

Academic Editor

PLOS ONE